

**THE ENEA-REG SYSTEM (v1.0), A MULTI-COMPONENT REGIONAL**
**EARTH SYSTEM MODEL. SENSITIVITY TO DIFFERENT**
**ATMOSPHERIC COMPONENTS OVER MED-CORDEX REGION**
Alessandro Anav[1], Adriana Carillo[1], Massimiliano Palma[1], Maria Vittoria Struglia[1], Ufuk Utku
Turuncoglu[2], Gianmaria Sannino[1]
[1] Italian National Agency for New Technologies, Energy and the Environment (ENEA), Rome,
Italy.
[2] National Center for Atmospheric Research, Boulder, CO, USA
**Abstract**
In this study, a new regional Earth system model is developed and applied to the Med-CORDEX
region. The ENEA-REG system is made up of two interchangeable regional climate models as
atmospheric components (RegCM and WRF), a river model (HD), and an ocean model
(MITgcm); processes taking place at the land surface are represented within the atmospheric
models with the possibility to use several land surface schemes of different complexity. The
coupling between these components is performed through the RegESM driver.
Here, we present and describe our regional Earth system model and evaluate its components
using a multidecadal hindcast simulation over the period 1980-2013 driven by ERA-INTERIM
reanalysis. We show how the atmospheric components are able to correctly reproduce both large-
scale and local features of the Euro-Mediterranean climate, although some remarkable biases are
relevant for some variables. In particular, WRF has a significant cold bias during winter over
North-Eastern bound of the domain, while RegCM systematically overestimates the wind speed
over the Mediterranean Sea. This latter bias has severe consequences on the ocean component:
we show that when WRF is used as the atmospheric component of the Earth system, the
performances of the ocean model are remarkably better compared with the RegCM version.
Our regional Earth system model allows studying the Euro-Mediterranean climate system and
can be applied to both hindcast and scenario simulations.



## 1.    Introduction

The Mediterranean basin is a complex region, characterized by the presence of pronounced topography and a complex land-sea distribution including a considerable number of islands and several straits. These features generate strong local atmosphere–sea interactions leading to the formation of intense local winds, like Mistral, Etesian and Bora which, in turn, dramatically affect the Mediterranean ocean circulation (e.g. Artale et al., 2010; Lebeaupin-Brossier et al. 2015; Turuncoglu and Sannino, 2017). Given the relatively fine spatial scales at which these processes take place, the Mediterranean basin provides a good opportunity to study regional climate, with a special focus on the air-sea coupling (Sevault et al., 2014; Turuncoglu and Sannino, 2017). For these reasons, regional coupled models have been developed and used to study both present and future Mediterranean climate system (e.g. Dubois et al., 2012; Ruti el al., 2016; Darmaraki et al., 2019; Parras-Berrocal et al., 2020); these models, depending on their complexity, include several physical components of the climate system, like atmosphere, ocean, land surface, rivers and biogeochemistry (both for land and ocean) (e.g. Drobinski et al., 2012; Sevault et al., 2014; Reale et al., 2020). Since the last two decades, an increasing number of studies have been performed over the Mediterranean basin and nowadays there is a coordinated effort for producing hindcast and future simulations over this region using regional coupled climate models sharing some common protocols (Ruti el al., 2016). In particular, the Coordinated Regional Climate Downscaling Experiment (CORDEX) was designed to produce, worldwide, high-resolution regional climate simulations through a coordinated experiment protocol ensuring that model simulations are carried out under similar conditions facilitating thus the analysis, intercomparison, and synthesis of different simulations (Giorgi et al., 2015; Giorgi et al., 2016). In the framework of the CORDEX program, regional climate model simulations dedicated to the Mediterranean  area  belong to the  Med-CORDEX initiative (Ruti el al., 2016, Somot et al., 2018).

From an atmospheric point of view, the Mediterranean region is a transition zone between arid subtropics and temperate mid-latitudes, characterized by low annual precipitation totals and high interannual variability; during winter, rain is brought by mid-latitude westerlies, while warm and dry summer results from the influence of subtropical remote forcing triggered by the Indian monsoon (Tuel and Eltahir, 2020). Future model projections have indicated that the


Mediterranean is expected to be one of the most prominent and vulnerable climate change "hotspots" in the world; in particular, a significant decline in the amount of precipitation is predicted by several models over the twenty-first century (Giorgi 2006; Tuel and Eltahir, 2020).

Given the complexity of the Mediterranean basin and the strong air–sea feedback, high resolution regional Earth system models are an optimal tool for accurate simulation of past, present and future climate over this region. The main aims of this paper are to present and evaluate the newly developed regional Earth system model ENEA-REG; in particular, we perform the evaluation run of the ENEA-REG system making a hindcast simulation using the ERA-interim reanalysis as boundary conditions. The performances of individual model components are evaluated comparing results with a wide range of observation-based datasets. Taking full advantage of the potential offered by the RegESM coupler (Turuncoglu 2019), that allows to build up in a modular way regional coupled models, the ENEA-REG is composed of two interchangeable regional climate models used as atmospheric components of the Earth system. Keeping fixed the ocean and rivers components, our model allows to explore the sensitivity of the ocean model to different atmospheric forcings: specifically, with the direct comparison of simulations differing for the atmospheric component, we infer the impact of different modeling choices on both air-sea processes and, consequently, on the ocean dynamics. Our results help to define possible future modelling strategies.

## 2.    Model description

### 2.1   The RegESM coupler

The ENEA-REG regional Earth system model has the capability to include several model components (atmosphere, river routing, ocean, wave) to allow different modeling applications.For each simulation, the components of the modeling system can be easily enabled or disabled via the driver's configuration file. In addition, the modeling framework also supports plugging new earth system sub-components (e.g. atmospheric chemistry, sea ice, ocean biogeochemistry) with minimal code changes through its simplified interface, which is called "cap". TheNational United Operational Prediction Capability (NUOPC) cap is a Fortran module that serves as interface to a model when it is used in a NUOPC-based coupled system;it is a small software layer that sits on top of a model code, making calls into it and exposing model data structures in a standard way (Turuncoglu, 2019).



In this study, the modeling system is configured to include three components: a regional
atmospheric climate model, a regional ocean model and an hydrological model. The driver used
to glue, regrid and exchange data among the three components of ENEA-REG modeling system
is RegESM (Turuncoglu 2019). The driver employs the Earth System Modeling Framework
(ESMF) library (version 7.1) and the NUOPC layer to connect and synchronize each model
component and perform interpolation among different horizontal grids (Turuncoglu 2019). While
the ESMF library deals with interpolation and regridding of exchanged fields, the NUOPC layer
simplifies common tasks of model coupling like component synchronization and run sequence
by providing additional wrapper layer between coupled model and ESMF framework
(Turuncoglu and Sannino, 2017; Turuncoglu 2019). It also allows defining different coupling
time intervals among the components to reproduce fast and slow interactions among the model
components (Turuncoglu and Sannino, 2017; Turuncoglu 2019). In this study, the model
coupling time step between ocean and atmosphere is set to 3-hours, while the coupling with the
hydrological model is defined as 1-day. In addition, the driver allows selecting the desired
exchange fields from a simple field database containing all available variables that can be
exported or imported by the different components. In this way, the coupled modeling system can
be easily adapted depending on the application and the particular configuration of the experiment
without any code customizations in both the driver and individual model components
(Turuncoglu, 2019).
In the experiment presented here, the atmospheric model retrieves sea surface temperature (SST)
from the ocean model (where grids are overlapped), while the ocean model collects surface
pressure, wind components, freshwater (evaporation-precipitation, i.e. E-P) and heat fluxes from
the atmospheric component. Similarly, the hydrological model uses surface and sub-surface
runoff simulated by the atmospheric component to compute the river drainage and exchanges
this field with the ocean component to close the water cycle. Further details on the ENEA-REG
framework and the interaction among the components are schematically depicted in **Figure 1**.
In the current work, we performed hindcast simulations covering the period 1st October 1979-31st
December 2013.

**2.2   The atmospheric components: WRF and RegCM**





The ENEA-REG regional Earth system model is made up of two interchangeable atmospheric
components: the Weather Research and Forecasting (WRF; Skamarock et al., 2008) model and
the REGional Climate Model (RegCM; Giorgi et al., 2012).
WRF is a limited-area, non-hydrostatic, terrain-following eta-coordinate mesoscale model
developed by the NCAR/MMM (National Center for Atmospheric Research, Mesoscale and
Microscale Meteorology division). WRF offers multiple options for various physical
parameterizations, thus it can be used to any region of the world for a wide range of applications
ranging from operational forecasts to realistic and idealized dynamical studies. In this work we
use the dynamical core ARW (Advanced Research WRF, version 3.8.1) (Skamarock et al.,
2008), with a single-moment 5 class scheme to resolve the microphysics (Hong et al., 2006) and
the Rapid Radiative Transfer Model for GCMs (RRTMG) for the shortwave and longwave
radiation (Iacono et al., 2008). Convective precipitation and cumulus parameterization are
resolved via the Kain-Fritsch scheme (Kain 2004), the planetary boundary layer (PBL) is
represented through the Yongsei University scheme (Hong et al., 2006), while the exchange of
heat, water and momentum between soil-vegetation and atmosphere is simulated by Noah–MP
land surface model(Niu et al, 2011). The model domain is projected on a Lambert conformal grid
with a horizontal resolution of 15 km and with 35 vertical levels extending from land surface up
to 50 hPa (**Figure 2a**). The initial and boundary meteorological conditions are provided by the
European Centre for Medium-Range Weather Forecast (ECMWF) reanalysis (Dee et al., 2011)
with a horizontal resolution of 0.75° every 6 h. The lateral buffer zone has a width of 10 grid
points and uses an exponential relaxation to provide the model with lateral boundary conditions.
In addition, we applied spectral nudging to temperature, wind components and moisture content
above the PBL; nudging is conducted every 6 h, consistent with the frequency of ERA-Interim
reanalysis data. A synthesis of parameterizations and input data used in this study is given in
**Table 1**.
The other supported atmospheric component of the regional Earth system model is RegCM
(version 4.5) a hydrostatic, compressible, sigma-p vertical coordinate model initially developed
by Giorgi (1990) and Giorgi et al. (1993a, 1993b) and then modified as discussed by Giorgi et al.
(2012); RegCM is maintained by ICTP's Earth System Physics (ESP) section. The dynamical
core of RegCM is based on the primitive equations, hydrostatic version of the National Centre
for Atmospheric Research (NCAR) and Pennsylvania State University mesoscale model MM5





(Grell et al., 1994). Similar to WRF, RegCM includes different physics and sub-grid
parameterization options. In this study, radiation is simulated with the radiative transfer scheme
of the global model CCM3 (Kiehl 1996), cumulus convection is resolved through the Grell
scheme (Grell 1993) with a Fritsch-Chappell scheme for unresolved convection, the planetary
boundary layer is represented viaa modified version of the Holtslag parameterization (Giorgi et
al 2012), while the exchange of heat, water and momentum between soil-vegetation and
atmosphere is simulated by the Biosphere-Atmosphere Transfer Scheme (BATS) (Dickinson et
al., 1993). The resolved scale precipitation is modeled with the SUBEX parameterization (Pal et
al, 2000).
The model domain (**Figure 2b**) is projected on a Lambert conformal grid with a horizontal
resolution of 20 km and with 23 vertical levels extending from land surface up to 50 hPa.
Similarly to WRF, we used ERA-Interim data to force RegCM and 6 grid-points in each side are
selected as relaxation zone with an exponentially decreasing relaxation coefficient (Giorgi et al.
1993) (**Table 1**).
A few modifications have been made both in WRF and RegCM to receive the oceanic surface
variables and send the atmospheric fields to the ocean component of the ENEA-REG system, as
described in **Figure 1**. Further details on the model's changes are described by Turuncoglu
171 (2019).


### 173  2.3  The ocean component: MITgcm

The ocean component of the ENEA-REG system is the Massachusetts Institute of Technology
General Circulation Model (MITgcm version c65; Marshall et al., 1997). The MITgcm solves
both the hydrostatic and nonhydrostatic Navier-Stokes equations under the Boussinesq
approximation for an incompressible fluid with a spatial finite-volume discretization on a
curvilinear computational grid using the z* rescaled height vertical coordinate (Adcroft and
Campin, 2004). MITgcm is designed to run on different platforms, from scalar to high-
performance computing (HPC) systems: it is parallelized via MPI through a horizontal domain
decomposition technique.
MITgcm is used by a broad community of researchers for a wide range of applications at various
spatial and temporal scales ranging from local/regional (e.g. Sannino et al., 2009; Furue et al.,
2015; Rosso et al., 2015; Sannino et al., 2015; McKiver et al., 2016; Sannino et al., 2017; Llasses



et al 2018; Peng et al., 2019) to global ocean simulations (e.g. Stammer et al., 2003; Forget et al.,
2015; Breitkreuz et al., 2018; Forget and Ferreira, 2019), including climate studies with MITgcm
coupled to atmosphere (e.g. Artale et al., 2010; Polkova et al., 2014; Sitz et al., 2017; Sun et al.,

188    2019).

In the configurations presented here, the MITgcm has been used in its hydrostatic, implicit free-
surface, partial step topography formulation (Adcroft et al, 1997) and has already been
customized and applied for simulating the Mediterranean circulation (Di Biagio et al 2019,
Cusinato et al. 2018). The model domain has a horizontal resolution of 1/12°, corresponding to
570x264 grid points, and covers the entire Mediterranean Sea with the boundary conditions in
the Atlantic Ocean (**Figure 2**). In the vertical the model is discretized using 75 unevenly spaced
Z-levels going from 1 m at the surface to about 300 m in the deepest part of the basin. We use
lateral open boundary conditions prescribed by the MITgcm Open Boundary Conditions (OBCS)
package. Temperature and salinity boundary conditions in the Atlantic Ocean are interpolated
from the global LEVITUS94 climatological monthly 3D data.
To ensure numerical stability a sponge layer is added to the open boundary of the domain. Each
variable is then relaxed toward the boundary values with a relaxation timescale that decreases
linearly with distance from the boundary. The thickness of the sponge layer in terms of grid
points is 18 and inner fields are relaxed toward boundary values using a 10 day period. Salinity
and temperature fields in the Mediterranean basin have been initialized using MEDATLAS/2002
climatology for the month of October. This month corresponds to a situation of stable vertical
stratification and can avoid sudden vertical mixing. A spin up procedure for the ocean model has
not been adopted, as in the regional ocean modeling community, the length of a spin-up is still a
matter of debate. Usually, for climate studies, long spin-up are desirable to avoid the models drift
considerably from the initial conditions and tend to converge toward a new state given by the
ocean physics (Sitz et al., 2017); as the aim of this study is the comparison of two coupled model
systems having in common the same ocean model, the MITgcm has the same initial and
boundary conditions in its two configurations.
Similar to the atmospheric models, we have modified the MITgcm model in order to be forced
by meteorological conditions derived by the atmospheric components of the ENEA-REG system
(see Turuncoglu and  Sannino 2017 for further details).
**2.4   The river routing model: HD**



The river discharge is a key variable in the Earth system modeling as it closes the water cycle
between the atmosphere and ocean. The ENEA-REG system uses the Hydrological Discharge
(HD, version 1.0.2) model, developed by the Max Planck Institute (Hagemann and Dümenil,
1998; Hagemann and Dümenil-Gates, 2001), to simulate freshwater fluxes over the land surface
and to provide a river discharge to the ocean model. The HD model uses a regular global grid
with a fixed horizontal resolution of 0.5° and it is forced by daily surface runoff and drainage
data. Similarly to other components, the HD model was slightly modified (Turuncoglu and
Sannino 2017) to retrieve surface runoff and drainage from the atmospheric components of the
regional coupled model and to provide the river discharge to the ocean component (**Figure 1**).
**3.       Experiment design and observational datasets**
In this work we present MED-CORDEX hindcast climate simulations performed with the
ENEA-REG model using both the atmospheric components of the system (i.e. WRF and
RegCM). Despite the simulations start time is October 1979, here we perform the model
validation over the period 1982-2013, using the first 2 years of simulation as spin up to initialize
all the fields of the different components of the coupled system. The validation of the coupled
model focuses on sea surface temperature, sea surface salinity and mixed layer depth for the
ocean, and 2m temperature, wind speed and freshwater and heat fluxes for the atmosphere. We
also compare river discharge from Po river as it influences the circulation of Adriatic Sea and the
formation of deep waters.
The simulated SST data are validated against the Objectively Interpolated Sea Surface
Temperatures (OISST v2, Reynolds et al., 2002, 2007), developed and distributed by the
National Oceanic and Atmospheric Administration (NOAA). The OISST composites
observations from different platforms (satellites, ships, buoys) on a 1/4° global grid and the gaps
are filled by interpolation (Reynolds et al., 2007).
Salinity data for the Mediterranean Sea are obtained from DIVA (data-interpolating variational
analysis); this tool allows to interpolate in situ observations to obtain gridded climatologies
(Brasseur et al., 1996).


For the mixed layer depth, we use a global climatology computed from more than one million
Argo profiles collected from 2000 to present (Holte et al., 2017); this climatology provides
estimates of monthly mixed layer depth on a global 1° gridded map.
As reference dataset to evaluate the performances of the atmospheric components of the ENEA-
REG system we use ERA5: this allows to test model's ability to reliably reproduce their parent
data (Mooney et al., 2013) and because, unlike other observational data, this dataset provides
information on both over land and ocean.
The observed river discharge of the Po river has been extracted from the series of measures at the
Ponte Lagoscuro station from the RivDIS dataset (Vorosmarty et al. 1998)
**4.      Results**
**4.1   Evaluation of atmospheric models**
The general ability of the atmospheric components of the ENEA-REG system to reproduce
realistic spatio-temporal patterns of the most relevant physical variables is assessed by
comparing model simulations with ERA5 during winter (DJF) and summer (JJA) seasons
averaged over the reference period 1982-2013. In the present analysis, in addition to spatial
patterns and anomalies maps, we also compute correlation patterns and domain-averaged bias to
provide a measure of the model's skills.
Looking at the surface air temperature (**Figure 3**), consistent with ERA5 data, during winter both
WRF and RegCM show a typical eastward gradient with temperature decreasing with increasing
continentally, while during summer the models correctly reproduce the decreasing south-north
gradient with colder areas localized over mountainous regions (i.e. Alps and Pyrenees). Looking
at the anomalies, WRF shows a remarkable cold bias during DJF over northeastern Europe, with
magnitudes larger than 4 °C. Such a cold bias over this region was already described in several
studies and it mainly depends on the choice of WRF physical parameterizations (e.g. Moonet et
al., 2013; Kotlarski et al., 2014; Katragkou et al., 2015). In a sensitivity study, where different
physical parameterizations schemes were used to represent radiation, microphysics, convection,
PBL and land surface, Mooney et al. (2013) reported that the simulated summer surface air
temperature is mostly controlled by the selection of land surface model, while during winter the
temperature shows some sensitivity to longwave radiation and very little sensitivity to other
parameterizations. Despite, when setting up WRF, we were aware of both the need to carefully



select parameterization combinations and the issues associated with some of the selected
parameterizations, we chose the present settings as they well reproduce wind fields over the
Mediterranean region, which is relevant when running WRF coupled with an ocean model.
Besides, as demonstrated by Mooney et al. (2013), over such a large domain, no single
combination of parameterizations yields optimal results. Unlike WRF, RegCM does not show
any remarkable bias during winter and, in general, it shows a cold bias ranging between 1 and 2
°C over the whole Mediterranean region. The good spatial agreement found during DJF between
the simulated surface air temperature and the reference data is confirmed by the high spatial
correlation varying between 0.98 in case of WRF to 0.99 for RegCM, while the domain-averaged
bias ranges from -1.3°C for WRF to -0.15°C for RegCM.
During summer, both WRF and RegCM show a similar bias pattern, with a warm bias extending
from France to Eastern Europe and reaching magnitudes of up to 3 °C in case of RegCM. This
result is consistent with Turuncoglu and Sannino (2017) who described a similar behaviour
running RegCM both standalone and coupled to ROMS ocean model, with a temperature
overestimation up to 2.0–2.5 °C during the summer season in central and eastern Europe.
Overall, our regional models well reproduce the observed spatial pattern, being the spatial
correlation larger than 0.99 for both WRF and RegCM. Considering the domain-averaged bias,
during JJA the configuration using WRF shows a slightly lower warm bias (0.1 °C) compared to
RegCM (0.14 °C).
Looking at precipitation, during winter both the ENEA-REG configurations have a good
agreement with ERA5 data, namely the atmospheric components are able to reproduce the major
precipitation maxima over the Alps, Balkans and western Norway with only a substantial local
dry bias in the areas around the coastlines of eastern Mediterranean. In contrast, during summer,
WRF and RegCM systematically simulate less precipitation over most of continental Europe,
with RegCM showing the largest dry bias (**Figure 4**). Interestingly, considering WRF, these
results are not consistent with Mooney et al. (2013), who reported a positive bias in mean daily
precipitation over Europe during summer and related this wet bias to the land surface scheme
used and partially to the microphysics scheme. However, Kotlarski et al. (2014) comparing three
WRF experiments showed a different sensitivity, with two simulations overestimating mean
summer precipitation and one underestimating it; they conclude that this result depends on the





choice of different microphysics schemes. On the other side, Turuncoglu and Sannino (2017)
found a similar bias pattern for RegCM during summer.
In general, the spatial performances of the ENEA-REG system are better when WRF is used as
the atmospheric component: the spatial correlation ranges between 0.97 during DJF to 0.95
during JJA, while the configuration with RegCM exhibits a slightly lower pattern correlation
(0.95 for DJF, 0.92 during JJA). Similarly, WRF has a smaller bias during summer (-0.42 vs -
0.54 mm/day), while during winter RegCM shows slightly better performances (-0.24 mm/day)
with respect to WRF (-0.27 mm/day); nevertheless, looking at **Figure 4** it should be noted that
the better performances of RegCM during winter are mainly explained by compensation between
dry and wet bias.
Despite the weak summer bias, the two atmospheric models well reproduce precipitation over the
sea, enhancing the reliability of freshwater flux exchanged with the ocean component of the
ENEA-REG system. Nevertheless, it should be noted that in the framework of coupled ocean-
atmosphere models, rather than precipitation, the water budget, defined as evaporation–
precipitation (E–P), plays a pivotal role in the dynamics of the ocean component. For this reason,
in **Figure 5** we show both the simulated inter-annual variability and mean seasonal cycle of the
area-averaged Mediterranean Sea precipitation, evaporation along with their difference (i.e. E-P).
Looking at precipitation, WRF shows a systematic dry bias over sea with respect to ERA5, while
RegCM is in good agreement with the reference value. The mean annual cycles suggest that
WRF underestimates rainfall during cold months (from November to March), while RegCM well
reproduces the observed seasonal cycle, with a weak overestimation between August and
October. Overall, the two configurations of ENEA-REG system well reproduce the reference
seasonal cycle, characterized by maximum values during fall and winter and minimum in
summer (JJA).
The total precipitation over the Mediterranean Sea is 409±41 mm/yr using WRF as atmospheric
component and 496±48 mm/yr in case of RegCM, while ERA5 predicts 469±50 mm/yr. In
general, these estimates agree with previous studies: in particular, in a different experiment,
where WRF was coupled with NEMO ocean model, Lebeaupin-Brossier et al. (2015) found a
precipitation budgets of 482±53 mm/yr over the period 1989–2008, concluding that this value is
in the upper part of the range given in the literature [290–510 mm/yr] (Mariotti et al. 2002;
Pettenuzzo et al. 2010; Romanou et al. 2010; Criado-Aldeanueva et al. 2012). Similarly, in a





regional climate system model developed over the Mediterranean Sea, where RegCM was
coupled with ROMS ocean model, Turuncoglu and Sannino (2017) found a mean annual
precipitation of 561 mm/yr during the temporal period 1988–2006; however, they also showed a
large variability in the estimates depending on the land-sea mask used to process data. In a
different configuration, where ALADIN climate model was coupled with NEMO ocean model,
Sevault et al. (2014) found a precipitation of 510 mm/yr over the time period 1980-2012, while
Sanchez-Gomez et al. (2011) compared 12 regional climate models finding a large spread among
models with mean annual precipitation estimates ranging between 347 and 606 mm/yr with a
mean value of 442±84 mm/yr.
Compared to ERA5, the evaporation is systematically overestimated by both RegCM and WRF
during our study period, despite the year-to-year variability is well reproduced and the mismatch
decreases with time (**Figure 5**); while in case of WRF this overestimation is mainly found
between April-September, RegCM overpredicts the evaporation during all months. Nevertheless,
the two configurations correctly reproduce the seasonal cycle, characterized by evaporation
minimum in May and maxima during late summer and winter months, when the gradient
between air–sea temperature is high and the wind speed is strong. The total evaporation over the
Mediterranean Sea is 1299±30 mm/yr and 1405±38 for WRF and RegCM, respectively, while
ERA5 has lower evaporation of 1198±59 mm/yr. Consistent with precipitation, our estimates
well agree with previous studies: Lebeaupin-Brossier et al. (2015) using WRF coupled to NEMO
found a total evaporation of 1442±45 mm/yr during the 1989–2008 period, while Turuncoglu
and Sannino (2017) using RegCM coupled to ROMS reported a value of 1388 mm/yr during the
1988–2006 period. Sevault et al. (2014) estimated a mean annual evaporation of 1390 mm/yr,
while Sanchez-Gomez et al. (2011) displayed a large variability among 12 regional climate
models, with annual mean estimates ranging between 1066 mm/year and 1618 mm/year, this
latter using RegCM offline forced by ERA40 data. The comparison with previous studies
highlights a general tendency of RegCM to overestimate the evaporation over the Mediterranean
sea, irrespective of the forcing data and parameterizations selected; this could be likely caused by
an overestimation of wind speed (discussed later).
Interestingly, because of bias compensation WRF and RegCM show a similar E-P estimate
(**Figure 5**); however, we found in both the configurations of the ENEA-REG system a
remarkable bias in E-P, with values larger than 100 mm/year, which could significantly affect the





ocean component. The monthly distribution of E–P shows, in both the ENEA-REG
configurations, a similar monthly distribution with ERA5 dataset with a peak in the late summer
caused by sparse precipitation and high evaporation. The total E-P estimated simulated using
WRF is 890±43 mm/yr while with RegCM we obtain a mean annual estimate of 909±45 mm/yr;
in contrast, ERA5 data has a lower E-P of 729±56 mm/yr.
In addition to freshwater flux, wind speed is also a key variable for ocean models as it controls
the evaporation over the sea surface and affects the ocean circulation through the drag stress.
**Figure 6** shows the near-surface wind speed as simulated by the ENEA-REG system and ERA5
reanalysis. The comparison with the observationally based dataset indicates that both WRF and
RegCM overestimate the wind speed over land during the two analyzed seasons, while over sea
the atmospheric models are able to correctly simulate the wind speed, especially over the Gulf of
Lion and the Aegean sea, where the structure and magnitude of dominant Mistral and Etesian
winds is well reproduced by WRF. In contrast, RegCM shows too weak Etesian during summer
and a general positive bias over the whole Mediterranean basin during DJF. This overestimation
by RegCM has a remarkable effect of deep water formation in the Levantine basin and affects
the deep convection and the mixed layer depth simulated by the ocean model (discussed later); in
addition, it is responsible for the large evaporative flux described in **Figure 5**.
It should also be noted that the large bias found over mountainous regions is clearly an artifact
due to the spatial resolution differences, with ERA5 reanalysis reproducing lower wind speed
than both WRF and RegCM because of its coarser resolution. In general, the two atmospheric
models have comparable performances in reproducing the observed spatial pattern; we find a
correlation of 0.98 for both models and seasons, except for RegCM during summer (0.97). In
contrast, WRF has a lower bias (1 m/s for DJF and 0.87 m/s for JJA) than RegCM (1.4 m/s for
DJF and 1.2 m/s for JJA). The higher agreement of WRF with ERA5 is a direct consequence of
the spectral nudging of wind data above the PBL.
Besides to freshwater flux and wind components, the surface net heat flux is used to drive the
ocean model of the ENEA-REG system (**Figure 1**); this variable represents the energy that the
ocean surface receives from the atmosphere and is computed from net longwave, net shortwave,
latent heat and sensible heat fluxes. Each component of the heat balance equation represents a
way ocean can gain or loss heat from the atmosphere: the latent heat flux controls the heat loss
by the ocean through evaporation, the sensible heat flux represents the heat loss by the ocean by





conduction to the atmosphere, the net shortwave radiation is the energy the ocean gains from the
Sun less a small amount of energy loss because of surface albedo, while the net longwave
radiation is the difference between the radiant energy emitted by the ocean and radiant energy
received from the atmosphere.
In **Figure 7** we compare the simulated net energy flux with ERA5 data; overall, the two
atmospheric models are in good agreement with the reference dataset during the analyzed
seasons, albeit a complex bias pattern is evident over the Mediterranean sea with WRF and
RegCM showing an interesting bias of opposite sign during summer and winter. The models
show similar skills in reproducing the ERA5 spatial patterns, both having a correlation of 0.96
during DJF, while in JJA RegCM (0.97) is slightly better than WRF (0.96); similarly, RegCM
also exhibits the lowest bias during both DJF (-1.3 $W/m^2$ vs 7.8 $W/m^2$) and JJA (3.1 $W/m^2$ vs
10.3 $W/m^2$). Looking at the spatial bias in more details, WRF shows a systematic positive bias
over the land surface up to 10 $W/m^2$ during winter and 15-20 $W/m^2$ in summer, while RegCM
well matches ERA5 data in DJF with bias lower than 5 $W/m^2$ but with a systematic negative bias
over the land ranging between -10 $W/m^2$ and -15 $W/m^2$ during JJA.
To further extend the analysis, in **Figure 8** we compare the monthly climatology of energy flux
components averaged over the whole Mediterranean Sea with ERA5 data. The analysis of model
results suggests that the latent heat is systematically overestimated by RegCM during the whole
year, whereas WRF is in good agreement with ERA5 during cold seasons (between October and
March) and it overestimates the latent heat flux in the remaining months (**Figure 8a**). The annual
mean estimates are 103±2.4 $W/m^2$ from WRF and 112±2.9 $W/m^2$ from RegCM, with ERA5
showing a slightly smaller flux (95±4.7 $W/m^2$). This result confirms previous findings about
RegCM, namely the too intense wind speed leads to a large latent heat flux and hence to an
overproduction of evaporative flux. In addition, our results are consistent with previous studies;
in particular, Turuncoglu and Sannino (2017) reported a value of 110.52 $W/m^2$ from RegCM
coupled to ROMS, whilst Sanchez-Gomez et al. (2011) showed a value of  128±5 $W/m^2$; in this
latter study, RegCM showed the largest overestimation of latent heat flux among 12 regional
climate models.
The sensible heat flux shows a similar behavior to that observed for the latent heat, namely
RegCM systematically overestimates this variable during the whole year, whilst WRF is closer to
the reference data (**Figure 8b**). The annual mean estimates are 12.9±1.2 $W/m^2$ from WRF,





17.6±1.2 W/m² from RegCM, while ERA5 has a slighter lower flux of 11.7±1.1 W/m².
Interestingly, using RegCM coupled to ROMS, Turuncoglu and Sannino (2017) found a smaller
sensible heat flux of 9.85 W/m², while Sanchez-Gomez et al. (2011) running RegCM offline
reported a value closer to our estimate (22±2 W/m²); as the sensible heat strictly depends on the
gradient between SST and air temperature the lower value of Turuncoglu and Sannino (2017)
could be explained by a large discrepancy between the SSTs simulated by the MITgcm and the
ROMS ocean models.
The mean annual cycle of net shortwave radiation is well simulated by the atmospheric models,
with WRF showing a perfect match compared to ERA5, while RegCM underestimates the
summer peak of about 25 W/m² and slightly overestimates the amount of radiation received by
the ocean from January to April (**Figure 8c**). The mean annual estimates are 199±1.2 W/m² form
WRF, 201±1.2 W/m² form RegCM and 198±1.1 W/m² form ERA5; for both the ENEA-REG
configurations, these estimates are in agreement with other studies (Sanchez-Gomez et al., 2011;
Turuncoglu and Sannino, 2017).
The comparison of simulated net longwave radiation with ERA5 data indicates that RegCM
underestimates the thermal radiation during the whole year, while WRF is in fair agreement
between March and October and overestimates the longwave radiation in the other months
(**Figure 8d**). In addition, the amplitude of seasonal variation is well captured by RegCM; in
contrast, WRF shows a stronger month-to-month variability. The mean annual net lonwave
radiation simulated by RegCM is -77.6±1.2 W/m², while WRF predicts -85.6±3.9 W/m² which is
very close to ERA5 dataset (-84.8±1.2 W/m²).

**4.2    Evaluation of ocean model**
**4.2.1. Surface processes**
The correct representation of physical processes taking place at the air-sea interface is crucial for
the success of a coupled climate simulation. A first evaluation of the goodness with which these
processes are simulated is given by the analysis of the ocean surface variables like Sea Surface
Temperature (SST) and Sea Surface Salinity (SSS).
**Figure 9** shows the comparison of simulated SST with OISST reference data. We recall that
SST, in a coupled simulation, is actually the same variable for ocean and atmosphere
components (where grids overlap), and guides the thermal exchange providing an active





feedback among the two components: the higher is the difference among SST and atmosphere
temperature, the larger will be the heat exchange at the interface that tends to lower such
difference. Looking at **Figure 9**, the coupled model well reproduces the OISST spatial pattern
with an agreement larger than 0.99 for both the configurations and seasons.WRF-MITgcm shows
moderate biases during winter (-0.24°C) and summer (0.23°C) while RegCM-MITgcm has a
widespread negative bias in winter (-0.9°C) and a positive bias in summer (0.25°C), with marked
spatial patterns in the eastern part of the Levantine Sea during winter and in the Sardinian Sea
during summer; it should be noted that the spatial average over the entire basin reduces the bias
within one degree, although the differences can be locally much more relevant, especially in the
RegCM-MITgcm configuration.
In spite of some large local bias, the RegCM-MITgcm well reproduces the observed interannual
variability, although it has a too marked year-to-year variability; in contrast, WRF-MITgcm well
captures the observed SST monthly anomalies (**Figure 10a**). Moreover, the WRF-MITgcm SST
seasonal cycle closely follows the reference dataset, while RegCM-MITgcm shows a
considerable SST underestimation between December and April and a slight overestimation in
the summer months (**Figure 10b**). Compared to similar modeling experiments, we note that an
overall cold bias is not unusual in coupled simulations of the Mediterranean Sea and the
magnitude of the biases obtained in the present study is comparable to the literature (Sevault et
al., 2014, Turuncoglu and Sannino, 2017, Reale et al.2020). In particular, the seasonal spatial
patterns in winter and summer closely resemble those shown in Turuncoglu and Sannino (2017),
although they used the ROMS model to simulate the Mediterranean Sea. More recently, Reale et
al. (2020) obtained a reduced  cold bias with respect to both the available literature and the
present experiment performed with RegCM-MITgcm; however a direct comparison is not
straightforward as their simulation period was limited to the years 1994-2006. Conversely,
considering WRF-MITgcm, our results are slightly better than similar simulations, being the bias
well below 0.3°C and no remarkable local bias are evident during the analyzed seasons.
Considering the SSS, compared to the reference data, both the simulations show very similar
spatial patterns and biases (**Figure 11**); we found the ocean model, in both its configurations,
saltier than the reference dataset, especially in the Adriatic Sea during summer. This is due to the
fact that the Adriatic Sea is a dilution basin, mainly because of the important freshwater supply
provided by rivers. In both the simulations the freshwater input from river runoff is heavily



underestimated by the interactive river routing model (**Figure 12**); this underestimation is more
evident in RegCM as a consequence of the larger drier precipitation bias found over land (**Figure**
**4**), resulting in a lower river baseline with respect to WRF (**Figure 12**).
Looking at the monthly SSS anomalies (**Figure 13a**) we found a similar temporal variability
compared to the reference data. Besides, the two configurations of the coupled model fairly
agree, although occasionally they are very different, as it happens in 1996, when WRF has an
remarkable drop in SSS due to the minimum in the freshwater flux (**Figure 5**) caused by
exceptional precipitation and river runoff during that year; interestingly, such a drop is also
evident in other observational datasets (Sevault et al., 2014).
Unlike the monthly SSS anomalies, the seasonal cycle of SSS for the two simulations is very
similar during all the months (**Figure 13b**), coherently with the freshwater flux seasonal cycle,
although both E and P over sea are more intense in RegCM than in WRF (**Figure 5**). Compared
to other studies, the mean bias of both WRF-MITgcm and RegCM-MITgcm is lower than that of
similar simulations for the Mediterranean Sea as it does not exceed 0.1 g/km on a basin mean
(e.g. Sevault et al., 2014, Turuncoglu and Sannino, 2017).

**4.2.2 Sea surface height and circulation**
The Strait of Gibraltar is the only connection between the Mediterranean basin and the Atlantic
Ocean. In general, the two-way exchange at the strait is constituted by an upper inflow of
Atlantic water and a lower outflow of relatively colder and saltier Mediterranean water.
However, the semidiurnal tidal effect is strong enough to reverse the direction of the flows
during part of the tidal cycle. As this exchange represents the main driver of the circulation in the
basin, the estimation of its value has been faced for decades.
The inflow transport derived from the two coupled simulations is about 1 Sv (**Table 2**);
similarly, the models predict a net transport of 0.06 Sv. Unfortunately, the estimate of the
transport obtained from the direct measurements of velocities is affected by the limited number
of moorings used that cannot resolve the structure of the entire section. Therefore, some
numerical models have also been used to reproduce and quantify the two way-exchange .
Estimates of mean inflow range from about 0.72 Sv of Bryden et al. (1994) to 1.68 Sv of
Bethoux (1979). Sannino et al. (2009) computed an inflow of 1.03 Sv using a three-dimensional





numerical model characterized by a very high resolution in the strait. Similarly, the long-term net
transport that balances the excess of evaporation over precipitation and river runoff in the
Mediterranean has a value of about 0.05 Sv (Bryden et al. 1994; Sannino et al., 2009);
noteworthy, our results well agree with these estimates (**Table 2**).
The Sicily strait connects the western and the eastern Mediterranean basins. The Modified
Atlantic Water (MAW) flows eastward in the upper layer and the Levantine Intermediate Water
(LIW) below it, in the opposite direction. Transports computed for this channel in the two
simulations are very close, with an eastward value of about 1.3 Sv and a net of a few hundredth
of Sv. These results are in agreement with the estimate of 1.1 Sv obtained in the experimental
work of Astraldi et al. (1999) and with the numerical model estimates ranging from 0.7Sv to 1.2
Sv (Fernandez et al. 2005, Zavatarelli & Mellor, 1995, Béranger et al. 2005).
The mean annual current velocity at 30 m depth and the mean annual Sea Surface Height (SSH)
are analyzed in **Figure 14** for WRF-MITgcm (a) and RegCM-MITgcm (b), respectively. The
two simulations depict a similar mean annual circulation, both at the surface (i.e SSH) and at the
intermediate level (i.e. velocities), with similar large-scale features.
The Atlantic Water (AW) circulation picture is in good agreement with those described by Millot
and Taupier-Letage (2005) and Pinardi et al. (2013), the first being mainly based on both in situ
and remotely sensed datasets, the latter resulting from a reanalysis performed with a model
having an horizontal resolution of 1/16° x 1/16°. In particular, Atlantic surface waters enter at
Gibraltar, are trapped into gyres in the Alboran Sea and then exit, dividing into two branches:
one sticking to the North-African coast, forming the Algerian current and the other in the
direction of the Balearic Islands. This latter detaches from the coast and flows south of Ibiza
Island generating an intense jet flowing eastward. This current receives the contribution of the
Southern edge of the Lion cyclonic gyre after the Balearic Sea and generates the Southern
Sardinian Current flowing along the west coast of Sardinia and merging with the Algerian
current. The Southern Sardinian Current branches in three parts (Béranger et al., 2004; Pinardi et
al., 2006): the southernmost branch produces the Sicily Strait Tunisian current, the central one
forms the Atlantic Ionian Stream (Robinson et al., 1999; Onken et al., 2003; Lermusiaux and
Robinson, 2001) and the northernmost one enters in the Tyrrhenian Sea giving rise to the South-
Western Tyrrhenian gyre. Finally, the Atlantic waters penetrate into the eastern basin through the
Sicily Strait. Noticeably, all these structures are very well defined in both the configurations of





the regional Earth system model (**Figure 14**). In addition, in the western Mediterranean basin,
the two model's versions show a wide cyclonic gyre, including the liguro-provencal current in
the Gulf of Lions.
The mean circulation in the Eastern basin is characterized by several features common to both
simulations. It is possible to appreciate how the surface water penetrates into the Adriatic Sea
with a cyclonic circulation, and it is possible to notice the presence of a counterclockwise
circulation in the Aegean Sea in both simulations; the WRF-based configuration is characterized
by a more intense eastward jet crossing the Eastern basin (**Figure 14**).
Also, the simulations reproduce quite clearly the places where deep water formation takes place:
the three cyclonic gyres located in the Gulf of Lyon, southern Adriatic Sea and in the Levantine
Sea. These cyclonic gyres concur with negative SSH values, which highlight the sinking of
surface waters.

**4.2.3 Heat and salt contents**
Mean annual temperature and salinity averaged over the entire Mediterranean basin and the
Western and Eastern sub-basins are shown in **Table 3**; here we present estimates from the DIVA
data, while for the two simulations we show the anomalies with respect to the reference data. The
average content of heat and salt has been computed over different vertical layers: the entire
column, the surface layer (0 -150m) corresponding approximately to the Atlantic Water, the
intermediate layer (150-600m) representing mainly the Levantine Intermediate Water, and the
deep layer (600-3500m) containing the Eastern and Western Mediterranean Deep Waters.
The average temperature of the whole water column, for each sub-basin, is in good agreement
with observations in both coupled runs, being the difference between modeled values and
observations not exceeding 0.2°C. Major discrepancies are concentrated in the upper layer of the
Eastern basin, where both models result colder than observation, with WRF-MITgcm showing an
underestimation of 0.45°C, while RegCM-MITgcm has a bias exceeding 1°C. Such discrepancy
reduces within the intermediate layer, while there is a slight overestimation in the deep layer, that
quite compensates for the error in the uppermost layers, when the total average is computed. In
the western basin the two models remain much closer to the observations, although RegCM-
MITgcm shows a systematic cold bias and WRF-MITgcm a systematic warm bias; however,
WRF is always closer to observations than RegCM. Notwithstanding the bias, we point out that



the mean values of the temperature within the different layers are compatible with those obtained
in analogous simulations, and are within the ensemble spread computed from the series of Med-
Cordex simulations analyzed by Llasses et al. (2018).
**Figure 15** shows the time series of mean annual temperature anomalies computed over the 1982-
2013 period for the surface and intermediate layers in the whole basin and in the Western and
Eastern sub-basins. Generally, the interannual variability of the whole basin is well captured by
the two simulations in both the surface layer and in the intermediate level. WRF-MITgcm is
remarkably close to observations between 0-150 m, while in the intermediate layer small
differences occur at the beginning of the simulations. The RegCM -MITgcm simulation shows a
slightly different behaviour with respect to observed data especially after the year 2006, when
the intensity of the positive anomalies is underestimated in both layers, although the year-to-year
variability is well reproduced. Altough the same general considerations hold for each of the two
sub-basins, we observe that WRF-MITgcm remarkably well captures the surface positive
anomaly in 1990 in the western basin, as well as the sequence of negative anomalies in the
eastern basin (1983,1987, and 1993). In the intermediate layer, the sudden drop of temperature
during 1993 is the signature of the Eastern Mediterranean Transient (EMT) phenomenon
(discussed in paragraph **4.2.4**).
The mean annual salinity averaged over the whole column (**Table 3**) is slightly overestimated in
both simulations (0.06 psu) mainly due to an overestimate of the salt content in the eastern sub-
basin. In the Eastern basin the maximum of salinity is correctly found in the intermediate layer
(150-600m), in correspondence of the LIW, although the RegCM-MITgcm simulation shows a
too slight decrease of the salinity from the intermediate to the deep layer. Such behaviour is
consistent with the higher values reached by the Mixed Layer Depth (MLD) in the same area
with respect to the MLD of the WRF-MITgcm simulation (discussed in paragraph **4.2.4**).
Similarly, in the western basin saltier intermediate water is clearly identified in the WRF run
with respect to RegCM, due to the combined effect of the advection of a saltier LIW and a less
intense deep convection, that in the western basin is mostly concentrated in the Gulf of Lion
area. The comparison of the MLD in the Gulf of Lion area (see paragraph **4.2.4**) supports this
hypothesis.
**Figure 16** shows the time series of mean annual temperature anomalies computed over the 1982-
2013 period for the surface and intermediate layers in the whole basin and in the Western and





Eastern sub-basins. While the entire basin variability is generally well reproduced, the behaviour
of models in the two sub-basins deserves some comment. In particular, in the western basin the
RegCM-MITgcm simulation fails in reproducing the drop in salinity of the uppermost layer
during the years 1990-1995. This is probably due to a too low freshwater flux in the RegCM-
MITgcm simulations in those years, confirmed by high values of the MLD. On the other hand, in
the eastern basin the WRF-MITgcm shows a freshwater anomaly in the 0-150m layer during the
years 1995-1997 that is not detectable in the reference data. However, it should be noted that the
same anomaly has also been observed in the SSS time series and is caused by exceptional
precipitation and river runoff as already reported by Sevault et al. (2014). Anyhow, such a drop
seems to affect mainly SSS and the surface layer, while it is scarcely transferred below 200 m. In
the intermediate layer both simulations show a steady increase in the salinity anomaly. RegCM-
MITgcm has almost a linear increase throughout the entire simulation period, due to the excess
of surface salinity and anomalous deep convection in the Levantine Sea, while WRF-MITgcm is
quite stable during the first half of the simulations and then shows a steep linear increase from
2000 onward.

### 631 4.2.4 Deep water formation

The formation of intermediate and deep waters due to sinking of dense water is one of the
fundamental processes taking place in the Mediterranean Sea, in both the Eastern and Western
sub-basins. Typical regions interested by this process are the Gulf of Lion, the South Adriatic,
the Cretan Sea and the Rhode Gyre. Such a process, mainly driven by the strong air-sea
interactions, takes place during the winter season, and is more effective during February. The
most active region for deep water formation is the Gulf of Lion, while intermediate and deep
waters are usually formed in the Adriatic and Levantine Sea, respectively.
The MLD is related to thermodynamic properties of seawater and is a pivotal variable helping in
the identification of deep-water formation events. High MLD values are related to strong air-sea
processes taking place at the surface or to preexisting stratification of the whole water column.
**Figure 17** compares the simulated monthly maximum MLD computed over most important
convective areas, i.e. the Levantine Sea, the Gulf of Lion, and the Adriatic Sea. Overall, RegCM-
MITgcm shows a more intense convection activity with respect to WRF-MITgcm, reaching the
deepest levels in all the analyzed regions. Looking at the Levantine region (**Figure 17a**), during





almost the entire simulation, the MLD simulated by RegCM-MITgcm exceeds 1000m depth, while in case of WRF-MITgcm, the MLD is more variable in time. The latter is often less than 1000m and reaches the entire water depth during a few events, which are well known and documented also in observations (Lascaratos et al. 1999; Malanotte et al., 1999; Roether et al., 2007). These events (1983,1987 and 1989), corresponding to intense atmospheric fluxes, have favoured the preconditioning of the eastern basin leading to the well-known phenomenon of the EMT. Therefore, we can conclude that the LIW formation is better reproduced in the simulation that has WRF as an atmospheric component.

Similarly, several MLD observation-based estimates are available in the Gulf of Lion for the period covered by our simulations (e.g. Martens and Schott 1998; Schroeder et al., 2008; Somot et al., 2016). Compared to these estimates, we observe that WRF-MITgcm simulation closely follows the timing of deep water formation in the Western Mediterranean, in particular the deep convection events of 1987 and 2005, with the exception of 1991 and 1992, identified by Somot et al. (2016) as years of intense mixing; in contrast, RegCM-MITgcm systematically presents a deeper MLD (**Figure 17b**).

In addition to the temporal evolution of MLD, in **Figure 18** we compare the mean spatial pattern of the MLD with ARGO data (Holte et al., 2017). Results suggest that the RegCM-MITgcm simulation not only reaches higher depths but also the downwelling regions are much more extended compared to both ARGO data and WRF-MITgcm simulation. This is particularly evident in the Levantine basin and, to a lesser extent, in the Western Mediterranean where the downwelling area extends from the Gulf of Lion to the Ligurian Sea.

The steady-state picture of the Mediterranean thermohaline circulation, in which the Eastern Mediterranean Deep Water (EMDW) is only of Adriatic origin, has been called into question by the discovery of the EMT. As described by many authors, there is observational evidence that during the '90s the main source of EMDW migrated to the Aegean Sea (Lascaratos et al., 1993; Malanotte et al., 1999; Wu et al., 2000; Roether et al., 2007; Beuvier et al., 2010). The common understanding is that the EMT has been the effect of many concurrent causes that make this process difficult to be simulated: the large heat loss from surface in the Levantine, the shifting from cyclonic to anticyclonic circulation in the Ionian that prevents the entering of freshwater in the Levantine basin, and the lower than usual freshwater flux from the Black Sea. Waters formed in the Aegean are warmer and saltier than that of the Eastern Mediterranean at the same levels,



and they are found at intermediate levels between LIW and EMWD of Adriatic origin. During the EMT period, instead, bottom levels were filled with newly formed waters of Aegean origin, while the less dense Adriatic waters were uplifted (Roether et al., 2007). All the studies agree on a massive dense-water formation in the Aegean Sea during the period 1987-1994 (e.g. Theocharis et al., 2002); as described by Theocharis et al. (1999), during the period 1986-1987, the Cretan Sea was characterized by a weak stratification. In the following years, water with densities higher than 29.2 was found at progressively upper layers in the Cretan Sea, with a significant formation rate in particular during 1989, due to an intrusion of deep waters from the central Aegean through the Myconos-Ikaria strait (Vervatis et al., 2013). Starting from 1989 dense water outflowed from the Cretan Arcs and was found in the Eastern Mediterranean Sea at levels between 700 and 1600 m. Then, dense water formation in the Cretan Sea increased during 1991 and 1992, the new water reached the upper layer of the Cretan basin, and the entire basin was filled with young water with density up to 29.3.

This phenomenon is remarkably well reproduced by the WRF-MITgcm simulation, both considering the timing of events and the density and volumes of newly formed waters, as shown in **Figure 19.** Here is depicted the volumes occupied by water with density higher than 29.2 kg/m3 and 29.3 kg/m3 in the Cretan Sea are; it can be seen that the period between 1983 and 1993 is characterized by an increase of the volume with three most significant peaks in 1984, 1989, and the highest in 1993, in both the simulations. Comparing with Sevault et al. (2014), the WRF-MITgcm has very similar behaviour with respect to both the timing of the events and the volumes formed, although they showed the whole Aegean Sea rather than the only Cretan Sea. In the 29.3 time series the event of 1993 is remarkably high, as expected, being this event the clear signature of the EMT. In contrast, RegCM-MITgcm is characterized by a more intense dense water formation, with the 29.2 water almost filling the Cretan basin during the whole simulation, while the 29.3 water almost fills the Cretan Sea in 1993, according to the EMT event. In the second part of the simulation, the volume of water filled with the densest water in the RegcM-MITgcm equals the volume occupied by the lower density water of WRF-MITgcm. Such an intense production of dense water in the Aegean Sea probably also affects the deep convection processes in the nearby Levantine Sea.





## 5.    Summary and conclusions

We presented a newly designed regional Earth system model used to study the climate variability
over the Euro-Mediterranean region. The performances of individual model components were
evaluated comparing results from the simulations with a wide range of observation-based
datasets.
Unlike other existing coupled atmosphere–ocean models, our system is made up of two
interchangeable atmospheric components (i.e. RegCM and WRF), offering thus the capability to
select the regional atmospheric model to be used. For each atmospheric configuration, we
performed a hindcast simulation over the period 1980-2013 using ERA-INTERIM reanalysis as
lateral boundary conditions.
Overall, results indicate that both RegCM and WRF correctly reproduce both large-scale and
local features of the Euro-Mediterranean climate, although some remarkable biases are relevant
for some variables. In particular, while WRF shows a significant cold bias during winter over
North-Eastern bound of the domain, RegCM systematically overestimates the wind speed over
the Mediterranean Sea.
Similarly, the ocean component correctly reproduces the analyzed surface ocean properties,
(along with their interannual variability) as well as the observed circulation in both the
configurations of the coupled model. Anyhow, results also point out remarkable better
performances when WRF is used to drive the ocean component of the coupled model; in fact,
because of the systematic overestimation of wind speed by RegCM, the ocean model has a cold
bias in SSTs during winter months and simulates a too deep mixed layer depth. This outcome is
mainly evident for the EMT, for which we showed that WRF-MITgcm is able to reproduce the
timing and the main characteristics of this event.
However, one could question that the overall better performances of WRF-MITgcm with respect
to RegCM-MITgcm could be attributable to the spectral nudging. This method allows the
passing of the driving model information not only onto the lateral boundaries but also into the
interior of the regional model domain (Waldron et al.1996; Heikkilä et al., 2011); this is achieved
by relaxing the model state towards the driving large-scale fields by adding a non-physical term
to the model equation (Omrani et al., 2015). Clearly, the spectral nudging allows a stronger
control by the driving forcing and thus a greater consistency between the regional model and
large-scale climate coming from the driving model. Nowadays, there is still some controversy on



the use of indiscriminate nudging in regional climate models (e.g. Omrani et al., 2015). Some
studies agree that nudging does not allow the regional model to deviate much from the driving
fields limiting the internal physics of the regional climate model (e.g. Sevault et al. 2014; Giorgi
2019). Considering the atmosphere-ocean coupling, Sevault et al. (2014) conclude that the use of
spectral nudging strongly constrains the synoptic chronology of the atmospheric flow and thus
the chronology of the air-sea fluxes and of the ocean response; they also found that this
facilitates day-to-day and interannual evaluation with respect to observations, but nudging also
limits the internal variability of the atmospheric component of the coupled model. Conversely, in
a different study on extreme events in the Mediterranean Sea performed with a coupled
atmosphere-ocean model, Lebeaupin-Brossier et al. (2015) found that nudging does not inhibit
small scale processes and thus potential air–sea feedbacks are still simulated. This result is
consistent with Omrani et al. (2015) who suggested that the spectral nudging technique does not
affect the small-scale fields since only the large scales are relaxed.
Anyhow, to evaluate the sensitivity of the modeled surface variables to nudging, we performed
the same simulation with WRF-MITgcm without nudging. Overall, results indicate that without
nudging WRF-MITgcm has poorer performances and in general is in agreement with RegCM-
MITgcm. For instance, considering the 2-m temperature over the Mediterranean sea, during DJF
the bias with nudging is -0.19°C, while without nudging becomes -0.6°C, closer to RegCM (-
0.96°C); similarly, during JJA the WRF bias increases significantly from 0.05 °C (with nudging)
to 0.95°C (without nudging), while RegCM has a bias of -0.76°C. Likewise, the performances of
the ocean model are strictly affected by the poorer performances of WRF without nudging:
looking at SST the bias during winter is doubled (-0.21°C vs -0.54°C) but still lower than
RegCM (-0.95°C), while during summer the performances of WRF-MITgcm without nudging
(0.8°C) are even worse than RegCM-MITgcm (0.27°C), so much poorer that the same
configuration using nudging (0.26°C).
This analysis reveals that spectral nudging helps to keep the large scale circulation of the
regional model in phase with the driving model; however, we remark that nudging  does not
avoid the model to develop large local bias related to poor representation of some processes; this
result is particularly clear for the cold bias during winter over North-Eastern bound of the
domain (**Figure 3**).





Notwithstanding the better performances, nudging has also to be used with caution: strong
inconsistencies between regional model and driving large-scale fields may lead to unrealistic
compensations within the model, for example, anomalous heat fluxes compensating for
temperature biases (Brune and Baehr, 2020).
We conclude that in the context of coupled atmosphere-ocean models, the correct representation
of surface winds is crucial to simulate ocean-atmosphere interactions correctly. In details, we
noted that poor representation of winds by RegCM led to significant deviations from
observations within the ocean model. This result is consistent with the poorer performances of
RegCM-MITgcm that mainly depend on the large bias in surface wind speed introduced with
RegCM. In this regard, as already discussed by Omrani et al. (2015), the wind above the PBL is
a key variable to nudge to simulate surface temperature, wind, and rainfall correctly. As wind
determines the transport of all conserved quantities like heat and moisture, their correct
representation has a relevant impact on several other quantities.
Finally, the comparison with offline results (not shown) suggests that atmosphere-ocean coupling
over the Mediterranean region remarkably changes the surface climate over the sea but, over
continental Europe, the climate is poorly constrained by the coupling. This is because the large-
scale systems mainly dominate the climate over central Europe originated in the Atlantic Ocean,
as already discussed in other studies (e.g. Somot et al., 2008; Artale et al. 2010; Turuncoglu and
Sannino, 2017). Nevertheless, as highlighted by Lebeaupin-Brossier et al. (2015), differences in
SST between offline and coupled simulations directly affect the local evaporation and
precipitation as well as the occurrences of extreme events.
Notwithstanding the low sensitivity of atmospheric components in the Mex-CORDEX region,
coupled models remain useful tools to predict future climate over the Mediterranean area (Artale
et al., 2010), which is widely recognized as climate change hot spot (e.g. Giorgi, 2006; Tuel and
Eltahir, 2020).

**Code availability**
The source code of the RegESM driver is distributed through the public code repository hosted
by GitHub (https://github.com/uturuncoglu/RegESM, last access: 24 December 2020). The
version that is used in this study is permanently archived on Zenodo and accessible under the
digital object identifier https://doi.org/10.5281/zenodo.4386712. The user guide and detailed



information about the modeling system and how to compile it are also distributed along with the
source code in the same code repository.
The standard version of WRF model is publicly available online at
https://github.com/NCAR/WRFV3/releases/tag/V3.8.1 (last access: 24 December 2020) but the
customized version that allow to couple with RegESM modeling system is permanently archived
on Zenodo and accessible under the digital object identifier
https://doi.org/10.5281/zenodo.4392230. The MITgcm model can be freely downloaded from its
web page (http://mitgcm.org/source-code/ (last access: 24 December 2020) but the substantially
modified version to allow coupling with RegESM modeling system can be accessible at
https://github.com/uturuncoglu/MITgcm and it is permanently archived on Zenodo and
accessible under the digital object identifier https://doi.org/10.5281/zenodo.4392260. The
RegCM model can be downloaded from public GitHub repository (https://github.com/ictp-
esp/RegCM, last access: 24 December 2020), while the HD model is available at
https://wiki.coast.hzg.de/display/HYD/The+HD+Model (last access: 24 December 2020) but
slightly customized version that enables coupling with RegESM modeling system can be
accessed from the public GitHub repository (https://github.com/uturuncoglu/HD) and it is
permanently archived on Zenodo and accessible under the digital object identifier
https://doi.org/10.5281/zenodo.4390527. For each model, the coupling support is provided
contacting the authors (alessandro.anav@enea.it; turuncu@ucar.edu;
gianmaria.sannino@enea.it).
The initial and boundary meteorological conditions, provided by the European Centre for
Medium-Range Weather Forecast (ECMWF), can be freely downloaded from the ECMWF web
page (https://apps.ecmwf.int/datasets/data/) after registration.
The LEVITUS94 monthly climatology for temperature and salinity is available at the web page
https://iridl.ldeo.columbia.edu/SOURCES/.LEVITUS94/.MONTHLY/ (last access: 24
December 2020). The Mediterranean and Black Sea database of temperature and salinity
(MEDATLAS/2002) is available at http://www.ifremer.fr/medar/.

**Author contributions**
UT wrote the RegESM driver, while all the authors worked on the coding tasks to couple the
model components through RegESM. AA and MS performed the simulations. All authors
discussed the results and contributed to the writing of the article.
**Competing interests**
The authors declare that they have no conflict of interest.
**Acknowledgements**
The computing resources and the related technical support used for this work have been provided
by CRESCO/ENEA-GRID High Performance Computing infrastructure and its staff

https://doi.org/10.5194/gmd-2020-248




(http://www.cresco.enea.it). CRESCO/ENEAGRID High Performance Computing infrastructure is funded by ENEA, the Italian National Agency for New Technologies, Energy and Sustainable Economic Development and by National and European research programs".

**Financial support**

This research has been supported by the SOCLIMACT project ("DownScaling CLImate imPACTs and decarbonisation pathways in EU islands, and enhancing socioeconomic and non-market evaluation of Climate Change for Europe, for 2050 and beyond"). UT is supported by the National Center for Atmospheric Research, which is a major facility sponsored by the National Science Foundation under Cooperative Agreement 1852977.

**References**

Adcroft, A., Hill, C., and Marshall, J.: Representation of topography by shaved cells in a height coordinate ocean model, Monthly Weather Review, 125, 2293-2315, 1997.

Adcroft, A., and Campin, J.-M.: Rescaled height coordinates for accurate representation of free-surface flows in ocean circulation models, Ocean Modelling, 7, 269-284, 2004.

Artale, V., Calmanti, S., Carillo, A., Dell'Aquila, A., Herrmann, M., Pisacane, G., Ruti, P. M., Sannino, G., Struglia, M. V., and Giorgi, F.: An atmosphere–ocean regional climate model for the Mediterranean area: assessment of a present climate simulation, Climate Dynamics, 35, 721-740, 2010.

Astraldi, M., Balopoulos, S., Candela, J., Font, J., Gacic, M., Gasparini, G., Manca, B., Theocharis, A., and Tintoré, J.: The role of straits and channels in understanding the characteristics of Mediterranean circulation, Progress in Oceanography, 44, 65-108, 1999.

Béranger, K., Mortier, L., Gasparini, G.-P., Gervasio, L., Astraldi, M., and Crépon, M.: The dynamics of the Sicily Strait: a comprehensive study from observations and models, Deep Sea Research Part II: Topical Studies in Oceanography, 51, 411-440, 2004.

Béranger, K., Mortier, L., and Crépon, M.: Seasonal variability of water transport through the Straits of Gibraltar, Sicily and Corsica, derived from a high-resolution model of the Mediterranean circulation, Progress in Oceanography, 66, 341-364, 2005.

Bethoux, J.: Budgets of the Mediterranean Sea. Their depenqance on the local climate and on the characteristics of the Atlantic waters, Oceanol. Acta, 2, 157-163, 1979.

Beuvier, J., Sevault, F., Herrmann, M., Kontoyiannis, H., Ludwig, W., Rixen, M., Stanev, E., Béranger, K., and Somot, S.: Modeling the Mediterranean Sea interannual variability during 1961–2000: focus on the Eastern Mediterranean Transient, Journal of Geophysical Research: Oceans, 115, 2010.

28





Brasseur, P., Beckers, J.-M., Brankart, J.-M., and Schoenauen, R.: Seasonal temperature and
salinity fields in the Mediterranean Sea: Climatological analyses of a historical data set, Deep
Sea Res. Part I, 43, 159– 192, 1996.
Breitkreuz, C., Paul, A., Kurahashi-Nakamura, T., Losch, M., and Schulz, M.: A dynamical
reconstruction of the global monthly mean oxygen isotopic composition of seawater, Journal of
Geophysical Research: Oceans, 123, 7206-7219, 2018.
Brossier, C. L., Bastin, S., Béranger, K., and Drobinski, P.: Regional mesoscale air–sea coupling
impacts and extreme meteorological events role on the Mediterranean Sea water budget, Climate
dynamics, 44, 1029-1051, 2015.
Brune, S., and Baehr, J.: Preserving the coupled atmosphere–ocean feedback in initializations of
decadal climate predictions, Wiley Interdisciplinary Reviews: Climate Change, 11, e637, 2020.
Bryden, H. L., Candela, J., and Kinder, T. H.: Exchange through the Strait of Gibraltar, Progress
in Oceanography, 33, 201-248, 1994.
Criado-Aldeanueva, F., Soto-Navarro, F. J., and García-Lafuente, J.: Seasonal and interannual
variability of surface heat and freshwater fluxes in the Mediterranean Sea: budgets and exchange
through the Strait of Gibraltar, International Journal of Climatology, 32, 286-302, 2012.
Darmaraki, S., Somot, S., Sevault, F., Nabat, P., Narvaez, W. D. C., Cavicchia, L., Djurdjevic,
V., Li, L., Sannino, G., and Sein, D. V.: Future evolution of marine heatwaves in the
Mediterranean Sea, Climate Dynamics, 53, 1371-1392, 2019.
Dee, D. P., Uppala, S. M., Simmons, A., Berrisford, P., Poli, P., Kobayashi, S., Andrae, U.,
Balmaseda, M., Balsamo, G., and Bauer, d. P.: The ERA-Interim reanalysis: Configuration and
performance of the data assimilation system, Quarterly Journal of the royal meteorological
society, 137, 553-597, 2011.
Dee, D. P., Uppala, S. M., Simmons, A., Berrisford, P., Poli, P., Kobayashi, S., Andrae, U.,
Balmaseda, M., Balsamo, G., and Bauer, d. P.: The ERA-Interim reanalysis: Configuration and
performance of the data assimilation system, Quarterly Journal of the royal meteorological
society, 137, 553-597, 2011.
Drobinski, P., Anav, A., Brossier, C. L., Samson, G., Stéfanon, M., Bastin, S., Baklouti, M.,
Béranger, K., Beuvier, J., and Bourdallé-Badie, R.: Model of the Regional Coupled Earth system
(MORCE): Application to process and climate studies in vulnerable regions, Environmental
Modelling & Software, 35, 1-18, 2012.
Dubois, C., Somot, S., Calmanti, S., Carillo, A., Déqué, M., Dell'Aquilla, A., Elizalde, A.,
Gualdi, S., Jacob, D., and L'hévéder, B.: Future projections of the surface heat and water budgets





of the Mediterranean Sea in an ensemble of coupled atmosphere–ocean regional climate models,
Climate dynamics, 39, 1859-1884, 2012.
Fernández, V., Dietrich, D. E., Haney, R. L., and Tintoré, J.: Mesoscale, seasonal and interannual
variability in the Mediterranean Sea using a numerical ocean model, Progress in Oceanography,
909    66, 321-340, 2005.

Forget, G., Campin, J.-M., Heimbach, P., Hill, C. N., Ponte, R. M., and Wunsch, C.: ECCO
version 4: An integrated framework for non-linear inverse modeling and global ocean state
estimation, 2015.
Forget, G., and Ferreira, D.: Global ocean heat transport dominated by heat export from the
tropical Pacific, Nature Geoscience, 12, 351-354, 2019.
Furue, R., Jia, Y., McCreary, J. P., Schneider, N., Richards, K. J., Müller, P., Cornuelle, B. D.,
Avellaneda, N. M., Stammer, D., and Liu, C.: Impacts of regional mixing on the temperature
structure of the equatorial Pacific Ocean. Part 1: Vertically uniform vertical diffusion, Ocean
Modelling, 91, 91-111, 2015.
Giorgi, F.: Simulation of regional climate using a limited area model nested in a general
circulation model, Journal of Climate, 3, 941-963, 1990.
Giorgi, F., Marinucci, M. R., and Bates, G. T.: Development of a second-generation regional
climate model (RegCM2). Part I: Boundary-layer and radiative transfer processes, Monthly
Weather Review, 121, 2794-2813, 1993.
Giorgi, F.: Climate change hot‐spots, Geophysical resear ch letters, 33, 2006.
Giorgi, F., Coppola, E., Solmon, F., Mariotti, L., Sylla, M., Bi, X., Elguindi, N., Diro, G., Nair,
V., and Giuliani, G.: RegCM4: model description and preliminary tests over multiple CORDEX
domains, Climate Research, 52, 7-29, 2012.
Giorgi, F., and Gutowski Jr, W. J.: Regional dynamical downscaling and the CORDEX initiative,
Annual Review of Environment and Resources, 40, 467-490, 2015.
Giorgi, F., and Gutowski, W. J.: Coordinated experiments for projections of regional climate
change, Current Climate Change Reports, 2, 202-210, 2016.
Giorgi, F.: Thirty years of regional climate modeling: where are we and where are we going
next?, Journal of Geophysical Research: Atmospheres, 124, 5696-5723, 2019.
Grell, G. A.: Prognostic evaluation of assumptions used by cumulus parameterizations, Monthly
weather review, 121, 764-787, 1993.





Grell, G. A., Dudhia, J., and Stauffer, D. R.: A description of the fifth-generation Penn
State/NCAR mesoscale model (MM5), 1994.
Hagemann, S., and Dümenil, L.: A parametrization of the lateral waterflow for the global scale,
Climate dynamics, 14, 17-31, 1997.
Hagemann, S., and Gates, L. D.: Validation of the hydrological cycle of ECMWF and NCEP
reanalyses using the MPI hydrological discharge model, Journal of Geophysical Research:
Atmospheres, 106, 1503-1510, 2001.
Heikkilä, U., Sandvik, A., and Sorteberg, A.: Dynamical downscaling of ERA-40 in complex
terrain using the WRF regional climate model, Climate dynamics, 37, 1551-1564, 2011.
Holte, J., Talley, L. D., Gilson, J., and Roemmich, D.: An Argo mixed layer climatology and
database, Geophysical Research Letters, 44, 5618-5626, 2017.
Hong, S.-Y., Dudhia, J., and Chen, S.-H.: A revised approach to ice microphysical processes for
the bulk parameterization of clouds and precipitation, Monthly weather review, 132, 103-120,
949 2004.

Hong, S.-Y., Noh, Y., and Dudhia, J.: A new vertical diffusion package with an explicit
treatment of entrainment processes, Monthly weather review, 134, 2318-2341, 2006.
Iacono, M. J., Delamere, J. S., Mlawer, E. J., Shephard, M. W., Clough, S. A., and Collins, W.
D.: Radiative forcing by long-lived greenhouse gases: Calculations with the AER radiative
transfer models, Journal of Geophysical Research: Atmospheres, 113, 2008.
Kain, J. S.: The Kain–Fritsch convective parameterization: an update, Journal of applied
meteorology, 43, 170-181, 2004.
Katragkou, E., García Díez, M., Vautard, R., Sobolowski, S. P., Zanis, P., Alexandri, G.,
Cardoso, R. M., Colette, A., Fernández Fernández, J., and Gobiet, A.: Regional climate hindcast
simulations within EURO-CORDEX: evaluation of a WRF multi-physics ensemble, 2015.
Kiehl, J., Hack, J., Bonan, G., Boville, B., and Briegleb, B.: Description of the NCAR
community climate model (CCM3). Technical Note, National Center for Atmospheric Research,
Boulder, CO (United States …, 1996.
Kotlarski, S., Keuler, K., Christensen, O. B., Colette, A., Déqué, M., Gobiet, A., Goergen, K.,
Jacob, D., Lüthi, D., and Van Meijgaard, E.: Regional climate modeling on European scales: a
joint standard evaluation of the EURO-CORDEX RCM ensemble, Geoscientific Model
Development, 7, 1297-1333, 2014.





Lascaratos, A., Williams, R. G., and Tragou, E.: A mixed-layer study of the formation of
Levantine Intermediate Water, Journal of Geophysical Research: Oceans, 98, 14739-14749,
969 1993.

Lascaratos, A., Roether, W., Nittis, K., and Klein, B.: Recent changes in deep water formation
and spreading in the eastern Mediterranean Sea: a review, Progress in oceanography, 44, 5-36,
972 1999.

Lebeaupin-Brossier, C., Bastin, S., Béranger, K., and Drobinski, P.: Regional mesoscale air–sea
coupling impacts and extreme meteorological events role on the Mediterranean Sea water
budget, Climate Dynamics, 44, 1029–1051, 2015.
Lermusiaux, P., and Robinson, A.: Features of dominant mesoscale variability, circulation
patterns and dynamics in the Strait of Sicily, Deep Sea Research Part I: Oceanographic Research
Papers, 48, 1953-1997, 2001.
Llasses, J., Jordà, G., Gomis, D., Adloff, F., Macías, D., Harzallah, A., Arsouze, T., Akthar, N.,
Li, L., and Elizalde, A.: Heat and salt redistribution within the Mediterranean Sea in the Med-
CORDEX model ensemble, Climate Dynamics, 51, 1119-1143, 2018.
Malanotte-Rizzoli, P., Manca, B. B., d'Alcala, M. R., Theocharis, A., Brenner, S., Budillon, G.,
and Ozsoy, E.: The Eastern Mediterranean in the 80s and in the 90s: the big transition in the
intermediate and deep circulations, Dynamics of Atmospheres and Oceans, 29, 365-395, 1999.
Mariotti, A., Struglia, M. V., Zeng, N., and Lau, K.: The hydrological cycle in the Mediterranean
region and implications for the water budget of the Mediterranean Sea, Journal of climate, 15,
987 1674-1690, 2002.

Marshall, J., Adcroft, A., Hill, C., Perelman, L., and Heisey, C.: A finite-volume,
incompressible Navier Stokes model for studies of the ocean on parallel computers, Journal of
Geophysical Research: Oceans, 102, 5753-5766, 1997.
Mertens, C., and Schott, F.: Interannual variability of deep-water formation in the Northwestern
Mediterranean, Journal of physical oceanography, 28, 1410-1424, 1998.
Millot, C., and Taupier-Letage, I.: Circulation in the Mediterranean sea, in: The Mediterranean
Sea, Springer, 29-66, 2005.
McKiver, W. J., Sannino, G., Braga, F., and Bellafiore, D.: Investigation of model capability in
capturing vertical hydrodynamic coastal processes: a case study in the north Adriatic Sea, Ocean
Sci., 12, 51-69, doi:10.5194/os-12-51-2016, 2016.
Mooney, P., Mulligan, F., and Fealy, R.: Evaluation of the sensitivity of the weather research and
forecasting model to parameterization schemes for regional climates of Europe over the period
1990–95, Journal of Climate, 26, 1002-1017, 2013.



Niu, G. Y., Yang, Z. L., Mitchell, K. E., Chen, F., Ek, M. B., Barlage, M., Kumar, A., Manning, K., Niyogi, D., and Rosero, E.: The community Noah land surface model with multiparameterization options (Noah-MP): 1. Model description and evaluation with local-scale measurements, Journal of Geophysical Research: Atmosp heres, 116, 2011.

Omrani, H., Drobinski, P., and Dubos, T.: Using nudging to improve global-regional dynamic consistency in limited-area climate modeling: What should we nudge?, Climate Dynamics, 44, 1627-1644, 2015.

Onken, R., Robinson, A. R., Lermusiaux, P. F., Haley, P. J., and Anderson, L. A.: Data-driven simulations of synoptic circulation and transports in the Tunisia-Sardinia- Sicily region, Journal of Geophysical Research: Oceans, 108, 2003.

Pal, J. S., Small, E. E., and Eltahir, E. A.: Simulation of regional-scale water and energy budgets: Representation of subgrid cloud and precipitation processes within RegCM, Journal of Geophysical Research: Atmospheres, 105, 29579-29594, 2000.

Parras-Berrocal, I. M., Vazquez, R., Cabos, W., Sein, D., Mañanes, R., Perez-Sanz, J., and Izquierdo, A.: The climate change signal in the Mediterranean Sea in a regionally coupled atmosphere–ocean model, Ocean Science, 16, 743-765, 2020.

Peng, Q., Xie, S.-P., Wang, D., Zheng, X.-T., and Zhang, H.: Coupled ocean-atmosphere dynamics of the 2017 extreme coastal El Niño, Nature communications, 10, 1-10, 2019.

Pettenuzzo, D., Large, W., and Pinardi, N.: On the corrections of ERA-40 surface flux products consistent with the Mediterranean heat and water budgets and the connection between basin surface total heat flux and NAO, Journal of Geophysical Research: Oceans, 115, 2010.

Pinardi, N., Arneri, E., Crise, A., Ravaioli, M., and Zavatarelli, M.: The physical, sedimentary and ecological structure and variability of shelf areas in the Mediterranean sea (27), The sea, 14, 1243-1330, 2006.

Pinardi, N., Zavatarelli, M., Adani, M., Coppini, G., Fratianni, C., Oddo, P., Simoncelli, S., Tonani, M., Lyubartsev, V., and Dobricic, S.: Mediterranean Sea large-scale low-frequency ocean variability and water mass formation rates from 1987 to 2007: A retrospective analysis, Progress in Oceanography, 132, 318-332, 2015.

Polkova, I., Köhl, A., and Stammer, D.: Impact of initialization procedures on the predictive skill of a coupled ocean–atmosphere model, Climate dynamics, 42, 3151-3169, 2014.

Reale, M., Giorgi, F., Solidoro, C., Di Biagio, V., Di Sante, F., Mariotti, L., Farneti, R., and Sannino, G.: The Regional Earth System Model RegCM-ES: Evaluation of the Mediterranean climate and marine biogeochemistry, Journal of Advances in Modeling Earth Systems, e2019MS001812,



Reynolds, R. W., Rayner, N. A., Smith, T. M., Stokes, D. C., and Wang, W.: An improved in situ
and satellite SST analysis for climate, Journal of climate, 15, 1609-1625, 2002.
Reynolds, R. W., Smith, T. M., Liu, C., Chelton, D. B., Casey, K. S., and Schlax, M. G.: Daily
high-resolution-blended analyses for sea surface temperature, Journal of Climate, 20, 5473-5496,
1039 2007.

Robinson, A., Sellschopp, J., Warn-Varnas, A., Leslie, W., Lozano, C., Haley Jr, P., Anderson,
L., and Lermusiaux, P.: The Atlantic ionian stream, Journal of Marine Systems, 20, 129-156,
1042 1999.

Roether, W., Manca, B. B., Klein, B., Bregant, D., Georgopoulos, D., Beitzel, V., Kovačević, V.,
and Luchetta, A.: Recent changes in eastern Mediterranean deep waters, Science, 271, 333-335,
1045 1996.

Roether, W., Klein, B., Manca, B. B., Theocharis, A., and Kioroglou, S.: Transient Eastern
Mediterranean deep waters in response to the massive dense-water output of the Aegean Sea in
the 1990s, Progress in Oceanography, 74, 540-571, 2007.
Romanou, A., Tselioudis, G., Zerefos, C., Clayson, C., Curry, J., and Andersson, A.:
Evaporation–precipitation variability over the Mediterranean and the Black Seas from satellite
and reanalysis estimates, Journal of Climate, 23, 5268-5287, 2010.
Rosso, I., Hogg, A. M., Kiss, A. E., and Gayen, B.: Topographic influence on submesoscale
dynamics in the Southern Ocean, Geophysical Research Letters, 42, 1139-1147, 2015.
Ruti, P. M., Somot, S., Giorgi, F., Dubois, C., Flaounas, E., Obermann, A., Dell'Aquila, A.,
Pisacane, G., Harzallah, A., and Lombardi, E.: MED-CORDEX initiative for Mediterranean
climate studies, Bulletin of the American Meteorological Society, 97, 1187-1208, 2016.
Sanchez-Gomez, E., Somot, S., Josey, S., Dubois, C., Elguindi, N., and Déqué, M.: Evaluation of
Mediterranean Sea water and heat budgets simulated by an ensemble of high resolution regional
climate models, Climate dynamics, 37, 2067-2086, 2011.
Sannino, G., Herrmann, M., Carillo, A., Rupolo, V., Ruggiero, V., Artale, V., and Heimbach, P.:
An eddy-permitting model of the Mediterranean Sea with a two-way grid refinement at the Strait
of Gibraltar, Ocean Modelling, 30, 56-72, 2009.
Sannino, G., Carillo, A., Pisacane, G., and Naranjo, C.: On the relevance of tidal forcing in
modelling the Mediterranean thermohaline circulation, Progress in oceanography, 134, 304-329,
1065 2015.

Sannino, G., Sözer, A. & Özsoy, E. A high-resolution modelling study of the Turkish Straits
System. Ocean Dynamics (2017) 67: 397. doi:10.1007/s10236-017-1039-2.





Schroeder, K., Ribotti, A., Borghini, M., Sorgente, R., Perilli, A., and Gasparini, G.: An extensive western Mediterranean deep water renewal between 2004 and 2006, Geophysical Research Letters, 35, 2008.

Sevault, F., Somot, S., Alias, A., Dubois, C., Lebeaupin-Brossier, C., Nabat, P., Adloff, F., Déqué, M., and Decharme, B.: A fully coupled Mediterranean regional climate system model: design and evaluation of the ocean component for the 1980–2012 period, Tellus A: Dynamic Meteorology and Oceanography, 66, 23967, 2014.

Sitz, L., Di Sante, F., Farneti, R., Fuentes-Franco, R., Coppola, E., Mariotti, L., Reale, M., Sannino, G., Barreiro, M., and Nogherotto, R.: Description and evaluation of the Earth System Regional Climate Model (RegCM-ES), Journal of Advances in Modeling Earth Systems, 9, 1863-1886, 2017.

Skamarock, W. C., and Klemp, J. B.: A time-split nonhydrostatic atmospheric model for weather research and forecasting applications, Journal of computational physics, 227, 3465-3485, 2008.

Somot, S., Sevault, F., Déqué, M., and Crépon, M.: 21st century climate change scenario for the Mediterranean using a coupled atmosphere–ocean regional climate model, Global and Planetary Change, 63, 112-126, 2008.

Somot, S., Houpert, L., Sevault, F., Testor, P., Bosse, A., Taupier-Letage, I., Bouin, M.-N., Waldman, R., Cassou, C., and Sanchez-Gomez, E.: Characterizing, modelling and understanding the climate variability of the deep water formation in the North-Western Mediterranean Sea, Climate Dynamics, 51, 1179-1210, 2018.

Somot, S., Ruti, P., Ahrens, B., Coppola, E., Jordà, G., Sannino, G., Solmon, F. Editorial for the Med-CORDEX special issue (2018) Climate Dynamics, 51 (3), pp. 771-777.

Stammer, D., Wunsch, C., Giering, R., Eckert, C., Heimbach, P., Marotzke, J., Adcroft, A., Hill, C., and Marshall, J.: Volume, heat, and freshwater transports of the global ocean circulation 1993–2000, estimated from a general circulation model constrained by World Ocean Circulation Experiment (WOCE) data, Journal of Geophysical Research: Oceans, 108, 7-1-7-23, 2003.

Sun, R., Subramanian, A. C., Miller, A. J., Mazloff, M. R., Hoteit, I., and Cornuelle, B. D.: SKRIPS v1. 0: A regional coupled ocean-atmosphere modeling framework (MITgcm-WRF) using ESMF/NUOPC, description and preliminary results for the Red Sea, 2019.

Theocharis, A., Nittis, K., Kontoyiannis, H., Papageorgiou, E., and Balopoulos, E.: Climatic changes in the Aegean Sea influence the Eastern Mediterranean thermohaline circulation (1986–1997), Geophysical Research Letters, 26, 1617-1620, 1999.

Theocharis, A., Klein, B., Nittis, K., and Roether, W.: Evolution and status of the Eastern Mediterranean Transient (1997–1999), Journal of Marine Systems, 33, 91-116, 2002.

 

Tuel, A., and Eltahir, E.: Why Is the Mediterranean a Climate Change Hot Spot?, Journal of
Climate, 33, 5829-5843, 2020.
Turuncoglu, U. U., and Sannino, G.: Validation of newly designed regional earth system model
(RegESM) for Mediterranean Basin, Climate dynamics, 48, 2919-2947, 2017.
Turuncoglu, U. U.: Toward modular in situ visualization in Earth system models: the regional
modeling system RegESM 1.1, Geoscientific Model Development, 12, 2019.
Vervatis, V. D., Sofianos, S. S., Skliris, N., Somot, S., Lascaratos, A., and Rixen, M.:
Mechanisms controlling the thermohaline circulation pattern variability in the Aegean–Levantine
region. A hindcast simulation (1960–2000) with an eddy resolving model, Deep Sea Research
Part I: Oceanographic Research Papers, 74, 82-97, 2013.
Waldron, K. M., Paegle, J., and Horel, J. D.: Sensitivity of a spectrally filtered and nudged
limited-area model to outer model options, Monthly weather review, 124, 529-547, 1996.
Wu, P., Haines, K., and Pinardi, N.: Toward an understanding of deep-water renewal in the
eastern Mediterranean, Journal of Physical Oceanography, 30, 443-458, 2000.
Zavatarielli, M., and Mellor, G. L.: A numerical study of the Mediterranean Sea circulation,
Journal of Physical Oceanography, 25, 1384-1414, 1995.








**TABLES**

**Table 1.** Set up of atmospheric components of the ENEA-REG system with main physical
parameterizations adopted in the simulations.

| Model set-up | WRF | RegCM |
|---|---|---|
| Domain | Med-CORDEX | Med-CORDEX |
| Simulation period | 1$^{st}$ October 1979-31$^{st}$ December 2013 | 1$^{st}$ October 1979-31$^{st}$ December 2013 |
| Horizontal resolution | 15 km | 20 km |
| Vertical resolution | 35 levels up to 50 hPa | 23 levels up to 50 hPa |
| Domain size | 350x280 (lon x lat) | 350x250 (lon x lat) |
| **Physical option** | **Adopted schemes** | **Adopted schemes** |
| Microphysics | WSM5 (single-moment 5 class) | SUBEX |
| Cumulus parameterization | Kain-Fritsch | Grell |
| Shortwave radiation | RRTMG | CCM3 |
| Longwave radiation | RRTMG | CCM3 |
| Land-surface | Noah-MP | BATS |
| Planetary boundary layer | Yonsei University Scheme | UW-PBL |
| Surface layer | Revised MM5 Monin-Obukhov scheme | Zeng |
| **Boundary condition** | **Configuration** | **Configuration** |
| Meteorological boundary | ERA-Interim (~75 km), 6h | ERA-Interim (~75 km), 6h |
| Relaxation zone | 10 points, exponential | 6 points, exponential |
| Nudging | Spectral | N/A |















**Table 2.** Mean annual water transport (in *Sv*) through the two main straits of Mediterranean Sea
over the period 1982–2013.

|  | Gibraltar | | | Sicily | | |
|---|---|---|---|---|---|---|
|  | Eastward | Westward | Net | Northward | Southward | Net |
| **WRF- MITgcm** | 0.965 | -0.905 | 0.061 | 1.332 | -1.357 | -0.025 |
| **RegCM- MITgcm** | 1.009 | -0.947 | 0.063 | 1.326 | -1.356 | -0.030 |

1141   .



**Table 3.** Averaged temperature (°*C*) and salinity (*psu*) at different depths for the DIVA dataset
and anomalies computed between the reference DIVA data and results from the coupled models.
Values are averaged over the entire Mediterranean Sea and over the western and eastern basins
for the temporal period 1982–2013.

| | | Temperature | | | | Salinity | | | |
|---|---|---|---|---|---|---|---|---|---|
| | | Depth [m] | | | | Depth [m] | | | |
| | | 0-150 | 150-600 | 600-3500 | 0-3500 | 0-150 | 150-600 | 600-3500 | 0-3500 |
| **MED** | **DIVA** | 16.20 | 14.04 | 13.33 | 13.78 | 38.43 | 38.73 | 38.62 | 38.63 |
| | **WRF** | -0.24 | 0.08 | 0.12 | 0.06 | -0.01 | 0.02 | 0.08 | 0.06 |
| | **RegCM** | -0.88 | -0.39 | 0.03 | -0.17 | -0.01 | -0.02 | 0.10 | 0.06 |
| **WMED** | **DIVA** | 14.99 | 13.42 | 12.98 | 13.26 | 37.95 | 38.51 | 38.47 | 38.43 |
| | **WRF** | 0.13 | 0.15 | 0.05 | 0.07 | -0.08 | -0.03 | 0.01 | -0.01 |
| | **RegCM** | -0.40 | -0.28 | -0.05 | -0.13 | -0.07 | -0.08 | 0.01 | -0.02 |
| **EMED** | **DIVA** | 16.89 | 14.41 | 13.56 | 14.10 | 38.70 | 38.86 | 38.73 | 38.75 |
| | **WRF** | -0.45 | -0.04 | 0.15 | 0.03 | 0.03 | 0.06 | 0.10 | 0.09 |
| | **RegCM** | -1.16 | -0.44 | 0.05 | -0.20 | 0.02 | 0.02 | 0.13 | 0.10 |















**FIGURES**


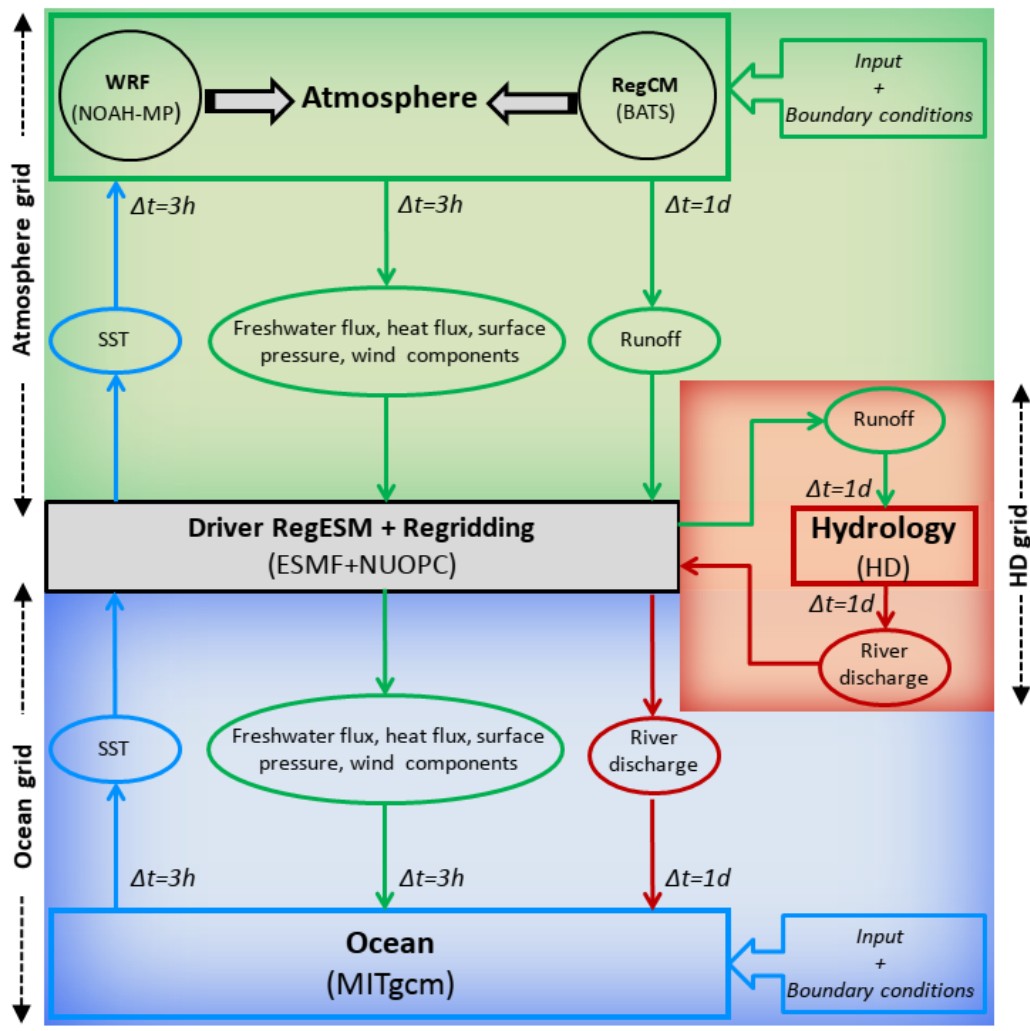


**Figure 1.** Schematic description of the ENEA-REG regional coupled model. The green block represents the atmosphere with the two components that can be selected and used (i.e. WRF and RegCM), the blue block is the ocean component (i.e. MITgcm), the red block represents the river routing component while the grey block is the ESMF/NUOPC coupler which collects, regrids and exchanges variables between the different components of the system.



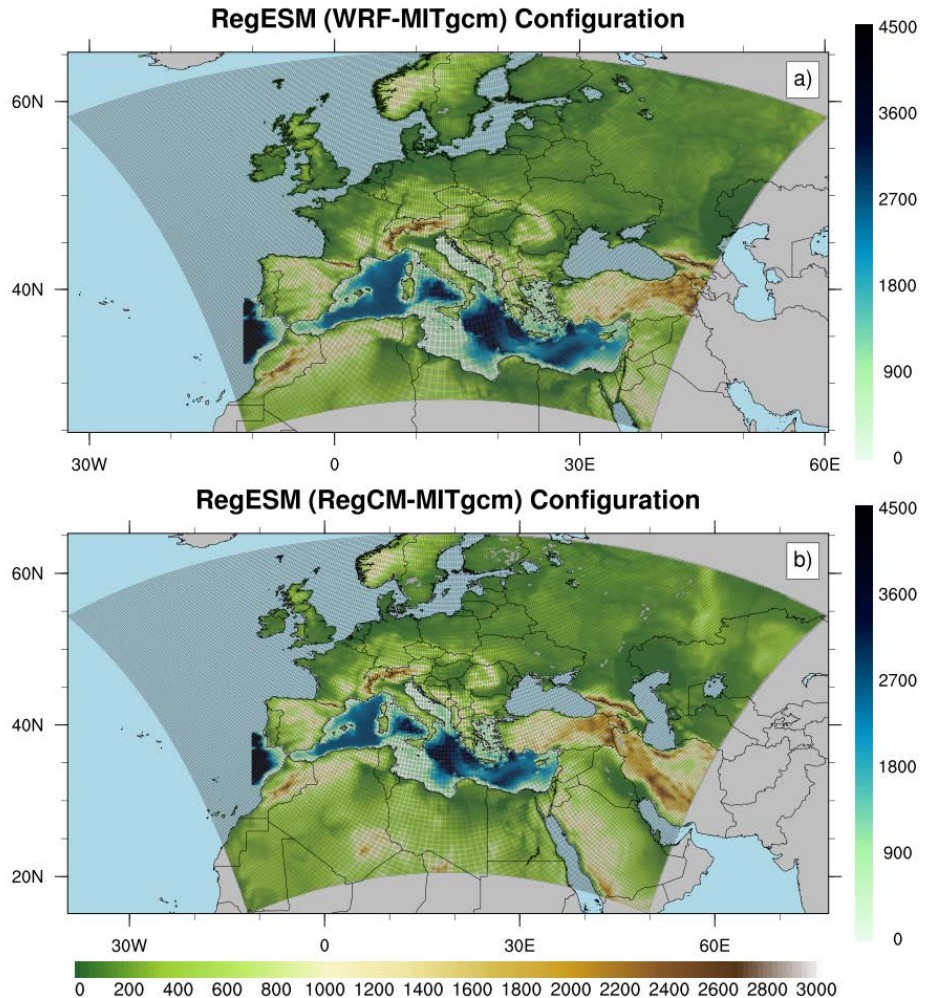

**Figure 2.** Different domains of the ENEA-REG system, with green shading representing the topography of the atmospheric models (i.e. WRF and RegCM, solid grey lines indicate the computational domain) and blue shading the bathymetry of the ocean component.


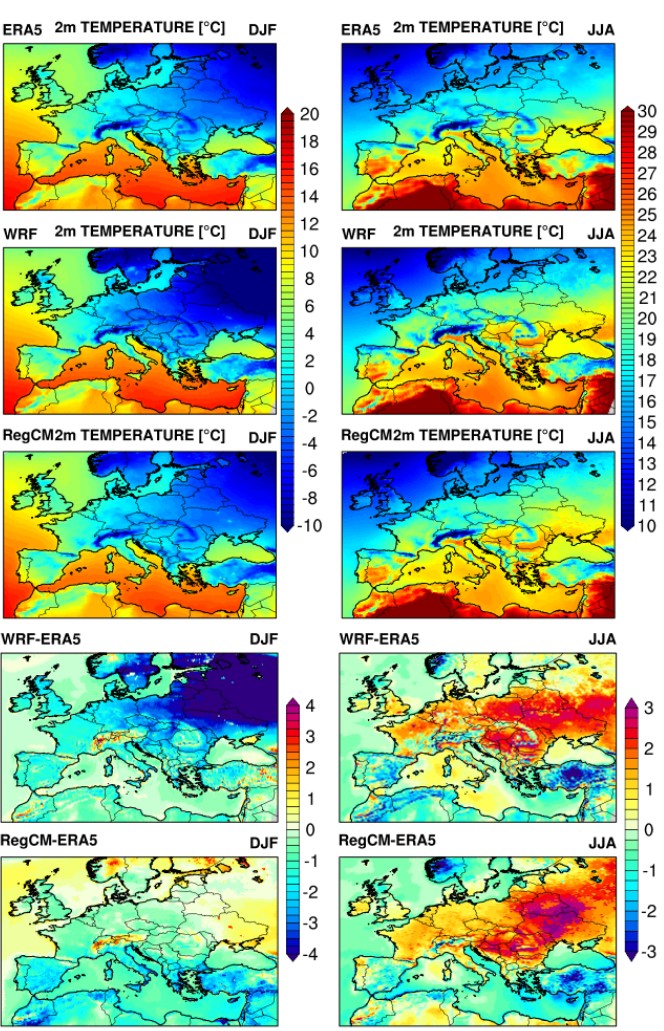

1205

**Figure 3.** Seasonal winter (DJF) and summer (JJA) spatial pattern (upper three panels) and bias
(lower two panels) of 2m air temperature as simulated by the coupled model using the two
atmospheric components (i.e. WRF and RegCM) and ERA5 dataset between 1982 and 2013.
Note that in the bias panels ERA5 data are interpolated into the atmospheric model grid. Mind
also the differences in colour scales between DJF and JJA climatologies.

1211

1212



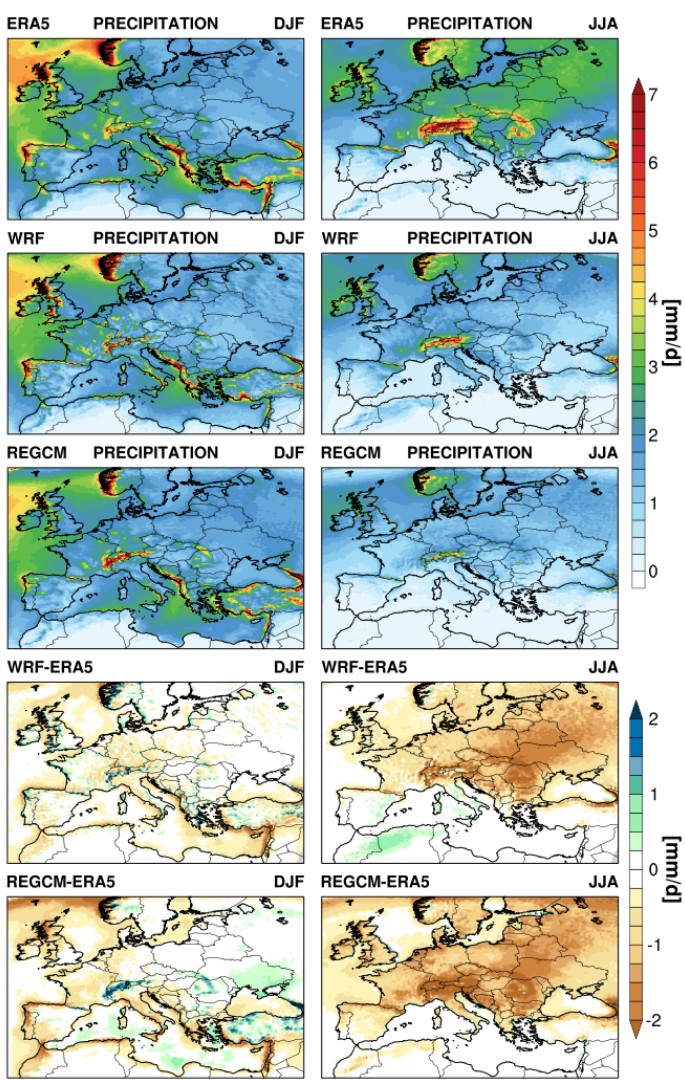

1213

**Figure 4.** Seasonal winter (DJF) and summer (JJA) spatial pattern (upper three panels) and bias
(lower two panels) of precipitation as simulated by the coupled model using the two atmospheric
components (i.e. WRF and RegCM) and ERA5 dataset between 1982 and 2013. Note that in the
bias panels ERA5 data are interpolated into the atmospheric model grid.

1218



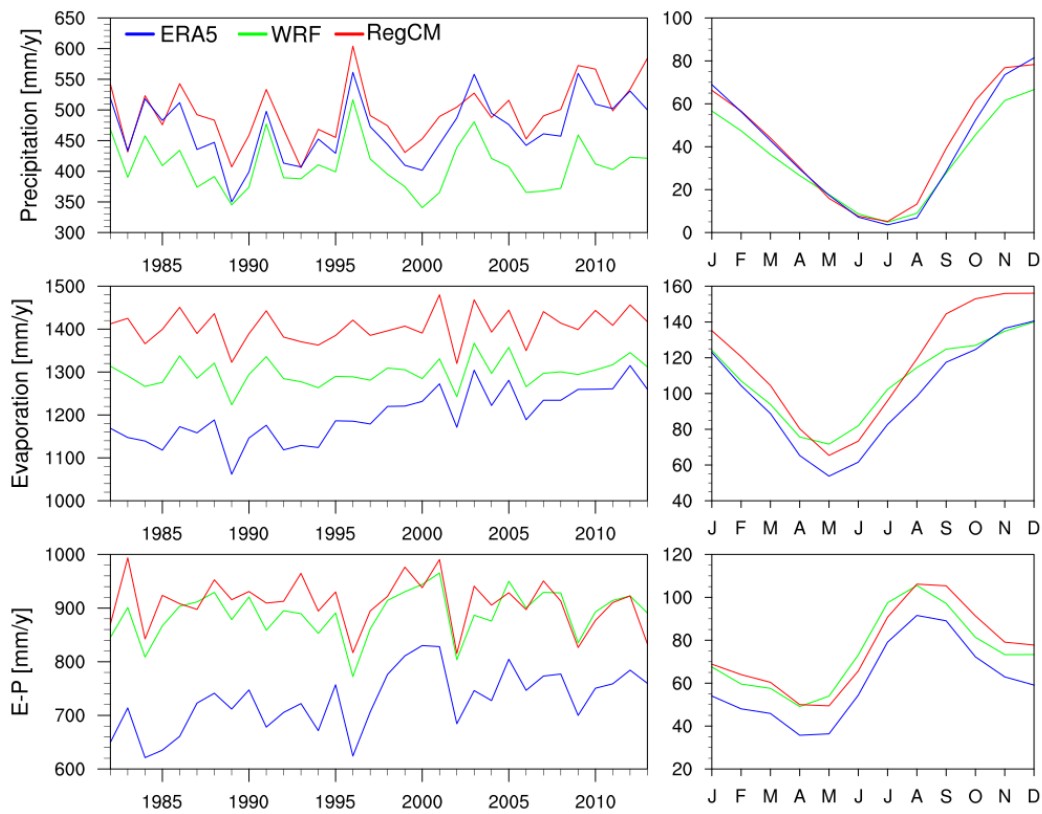

**Figure 5.** Interannual variability (left panels) and mean seasonal cycle (right panels, units mm/month) of freshwater flux components. i.e. precipitation (P), evaporation (E) and their difference (E-P), computed over the Mediterranean basin as simulated by the ENEA-REG system and ERA5 reanalysis.



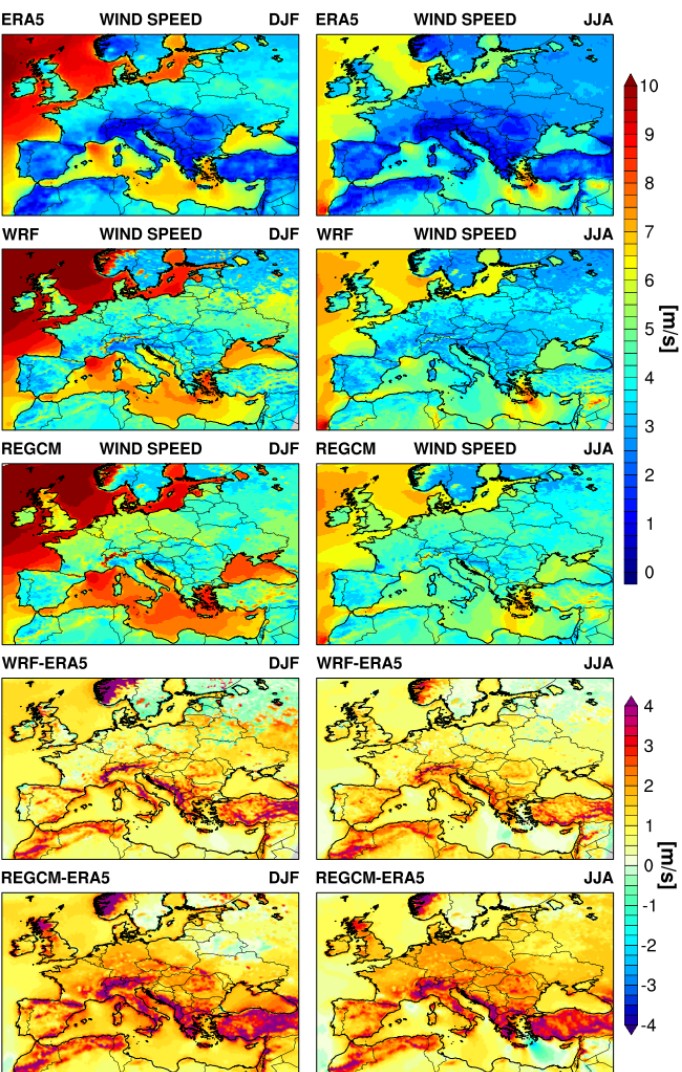

1224

**Figure 6.**Seasonal winter (DJF) and summer (JJA) spatial pattern (upper three panels) and bias (lower two panels) of 10m wind speed as simulated by the coupled model using the two atmospheric components (i.e. WRF and RegCM) and ERA5 dataset between 1982 and 2013. Note that in the bias panels ERA5 data are interpolated into the atmospheric model grid.

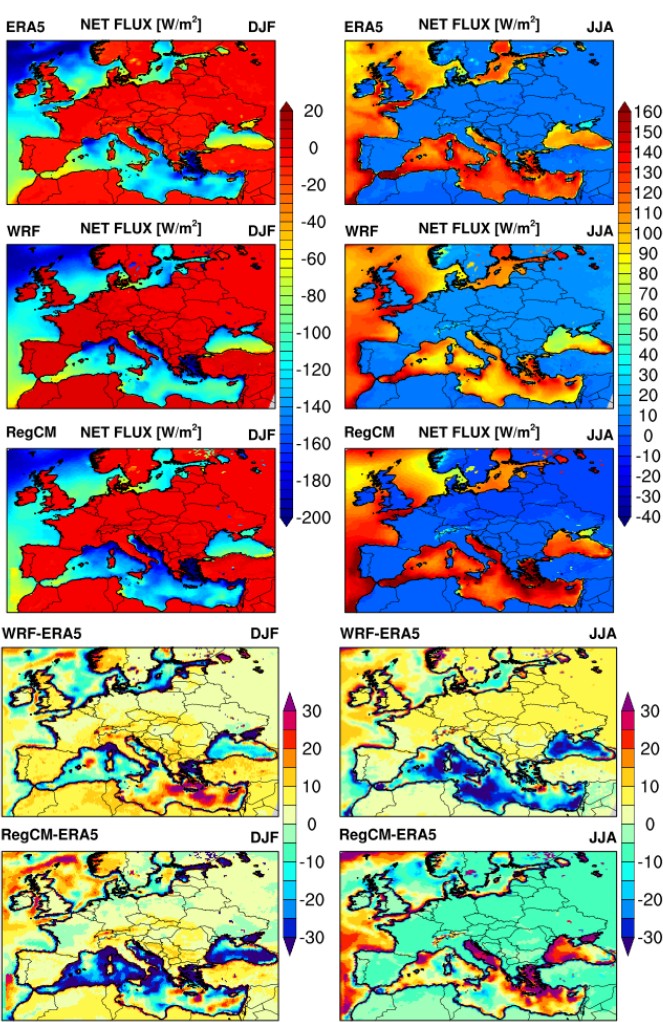

**Figure 7.** Seasonal winter (DJF) and summer (JJA) spatial pattern (upper three panels) and bias (lower two panels) of net heat flux as simulated by the coupled model using the two atmospheric components (i.e. WRF and RegCM) and ERA5 dataset between 1982 and 2013. Note that ERA5 data are interpolated into atmospheric model grids for comparison purposes. Mind also the differences in colour scales between DJF and JJA climatologies.

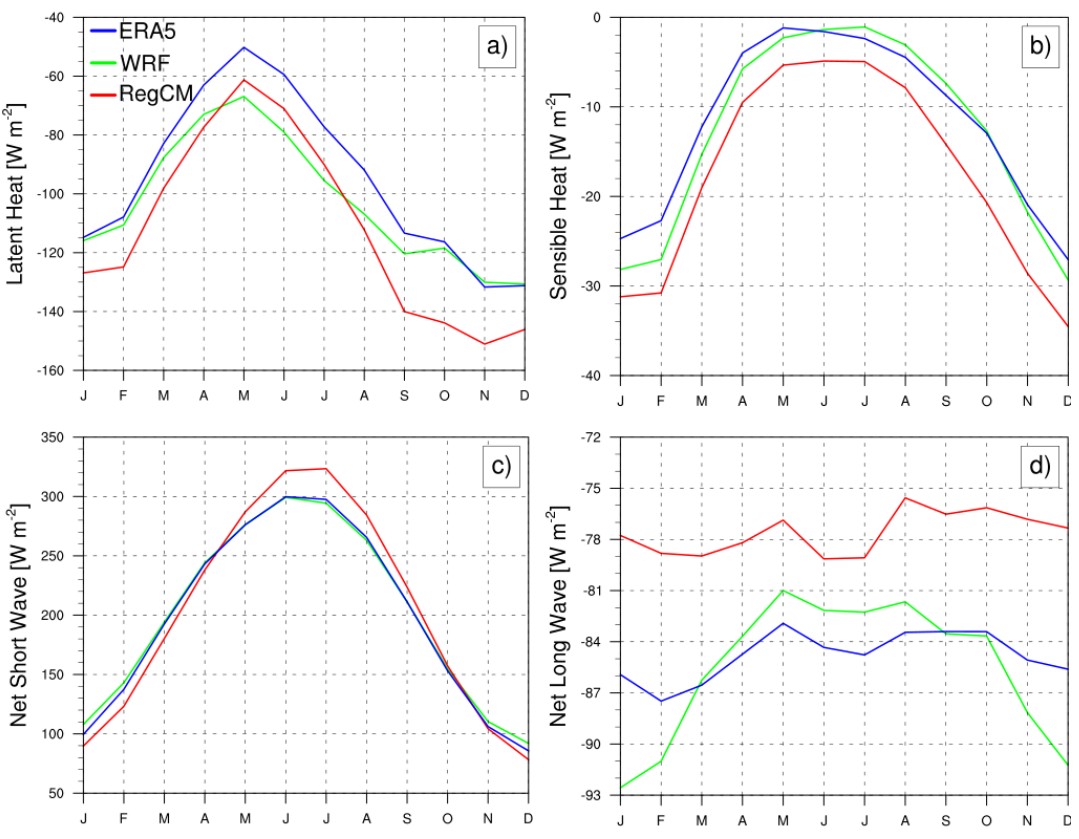

1236

**Figure 8.** Mean seasonal cycle of net heat flux components over the Mediterranean basin as simulated by the ENEA-REG system and ERA5 reanalysis over the period 1982-2013.





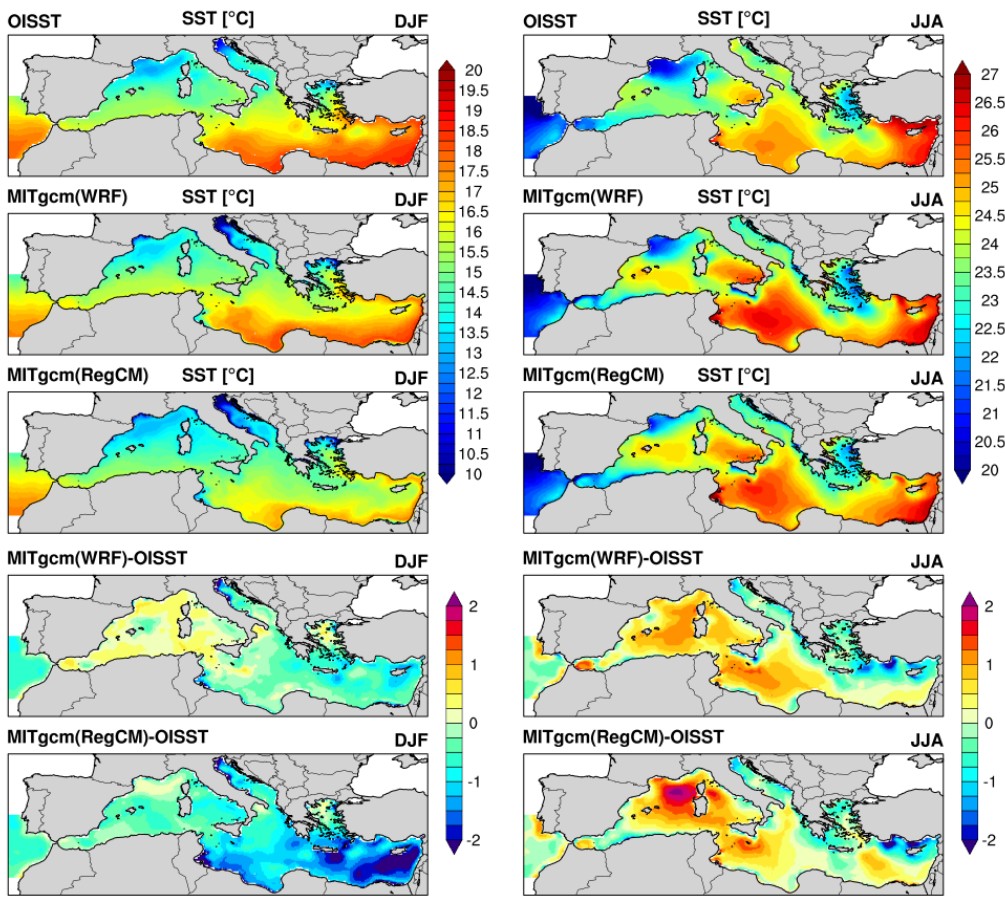

1239

**Figure 9.** Seasonal winter (DJF) and summer (JJA) spatial pattern (upper three panels) and bias (lower two panels) of sea surface temperature (SST [°C]) as simulated by the coupled model using the two atmospheric components as forcing (i.e. WRF and RegCM) and OISST dataset between 1982 and 2013. Note that OISST data are interpolated into ocean model grid for comparison purposes. Mind also the differences in colour scales between DJF and JJA climatologies.

1246



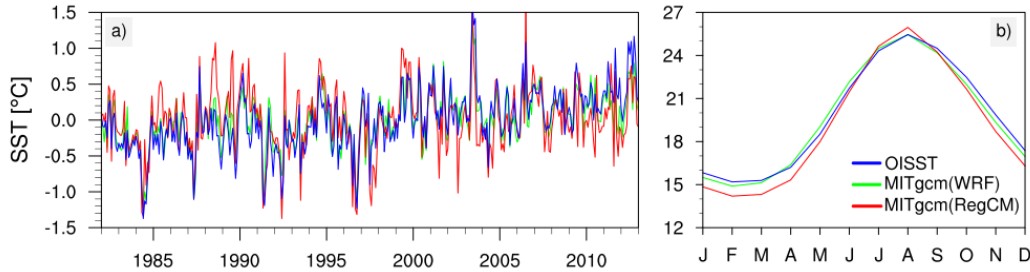

**Figure 10.** Comparison of monthly anomalies (left panel) and mean seasonal cycles (right panel) of sea surface temperature simulated by the ENEA-REG system with OISST observation.





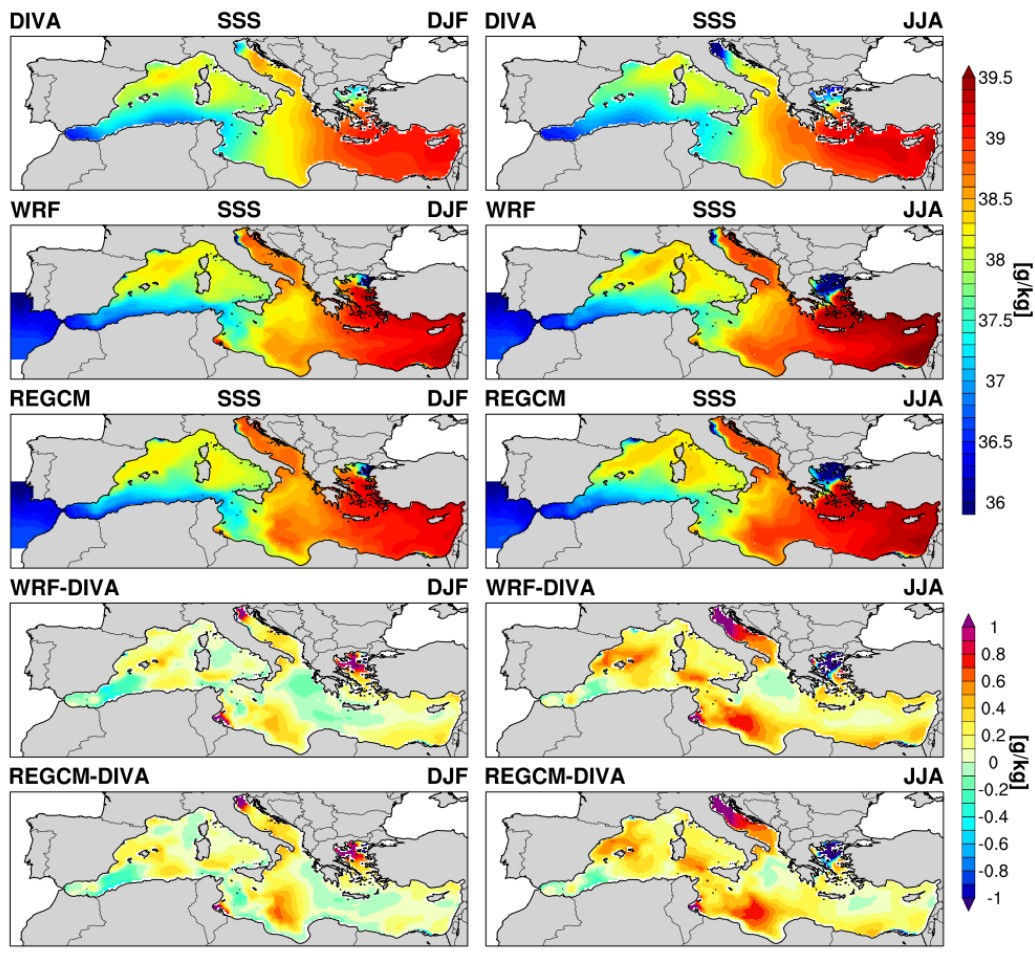

1251

**Figure 11.** Seasonal winter (DJF) and summer (JJA) spatial pattern (upper three panels) and bias (lower two panels) of sea surface salinity (SSS [*g/kg*]) as simulated by the coupled model using the two atmospheric components (i.e. WRF and RegCM) and DIVA dataset between 1982 and 2013. Note that DIVA data are interpolated into ocean model grid for comparison purposes.








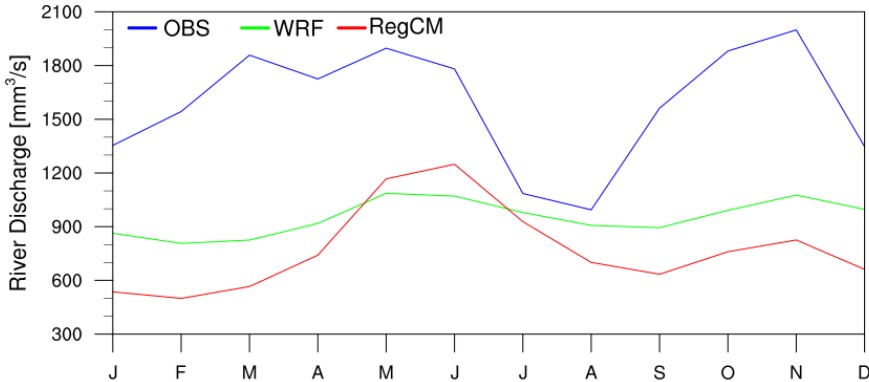


**Figure 12.** Mean seasonal cycle of the river discharge of the Po river into the Adriatic Sea as
simulated by the two configurations of the coupled model and the observational dataset RivDis.



















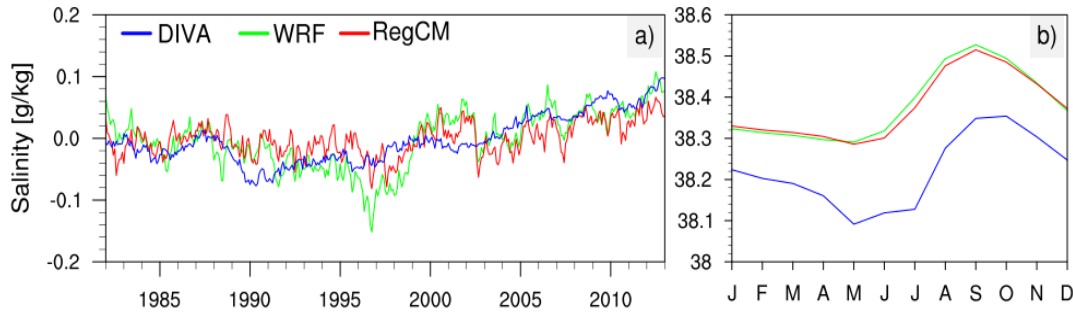


**Figure 13.** Comparison of monthly anomalies (left panel) and mean seasonal cycles (right panel) of sea surface salinity simulated by the ENEA-REG system with DIVA dataset.





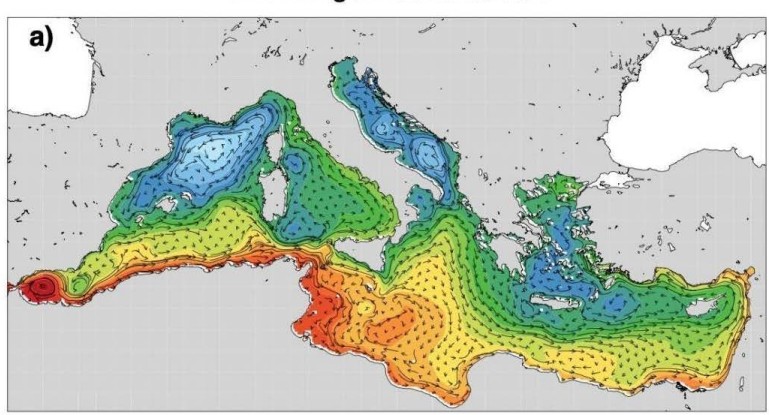

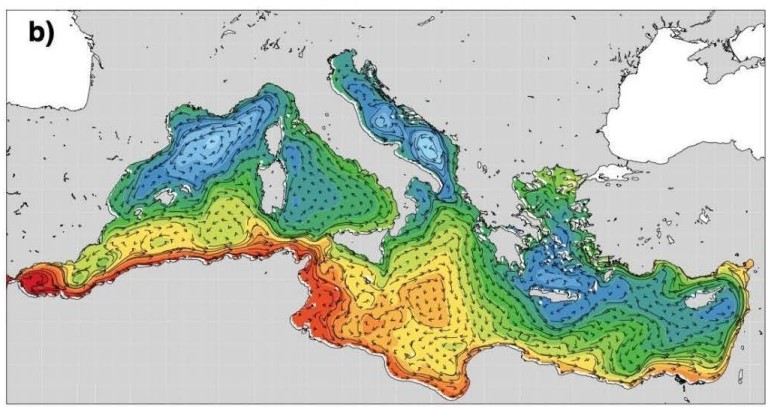

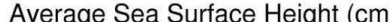

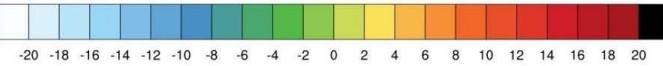

**Figure 14.** Mean annual sea surface elevation along with sub-surface (30m) circulation as simulated by the two configurations of the coupled atmosphere-ocean model; data are averaged over the temporal period 1982-2013.

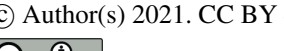



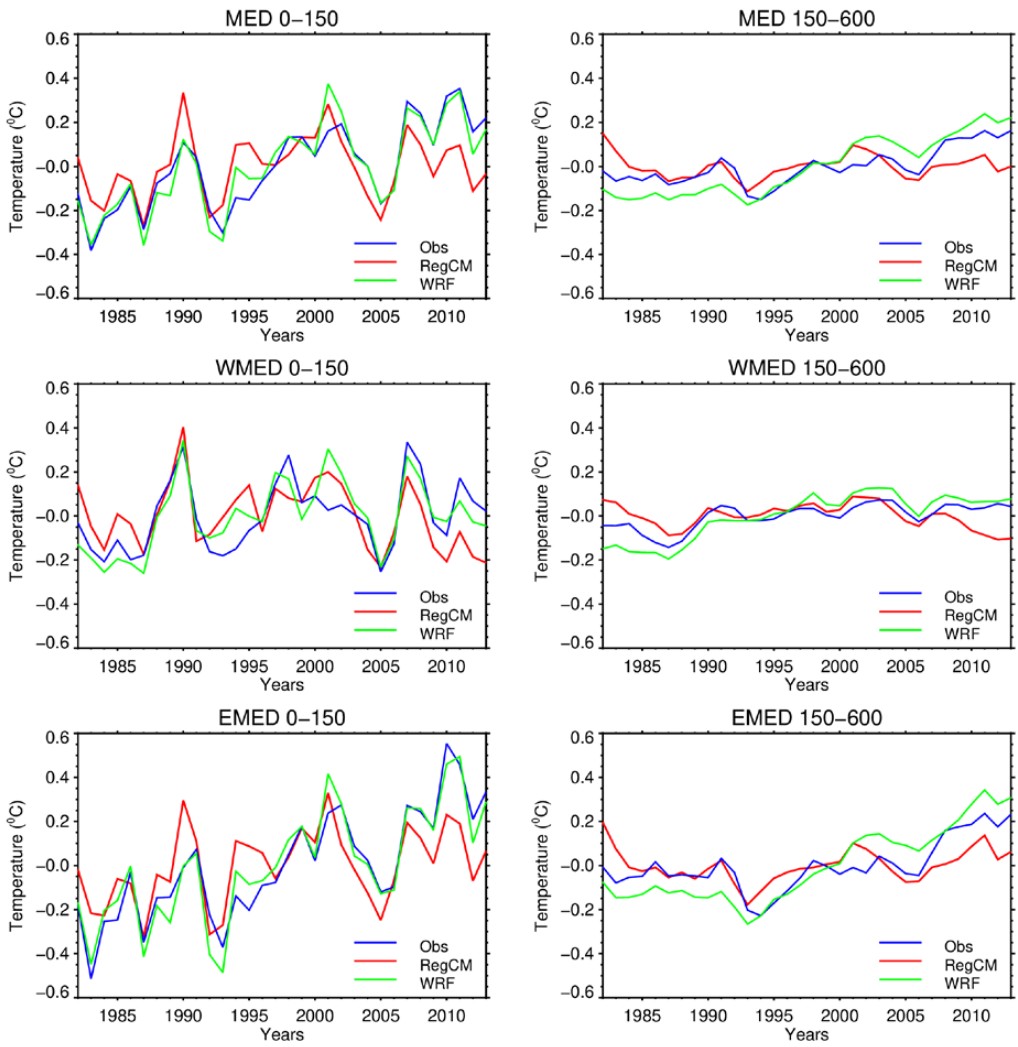

**Figure 15.** Annual mean temperature anomalies ($^{\circ}C$) for upper (0-150 m) and intermediate (150-
600 m) layers of the Mediterranean Sea, Western and Eastern basins over the period 1982-2013.



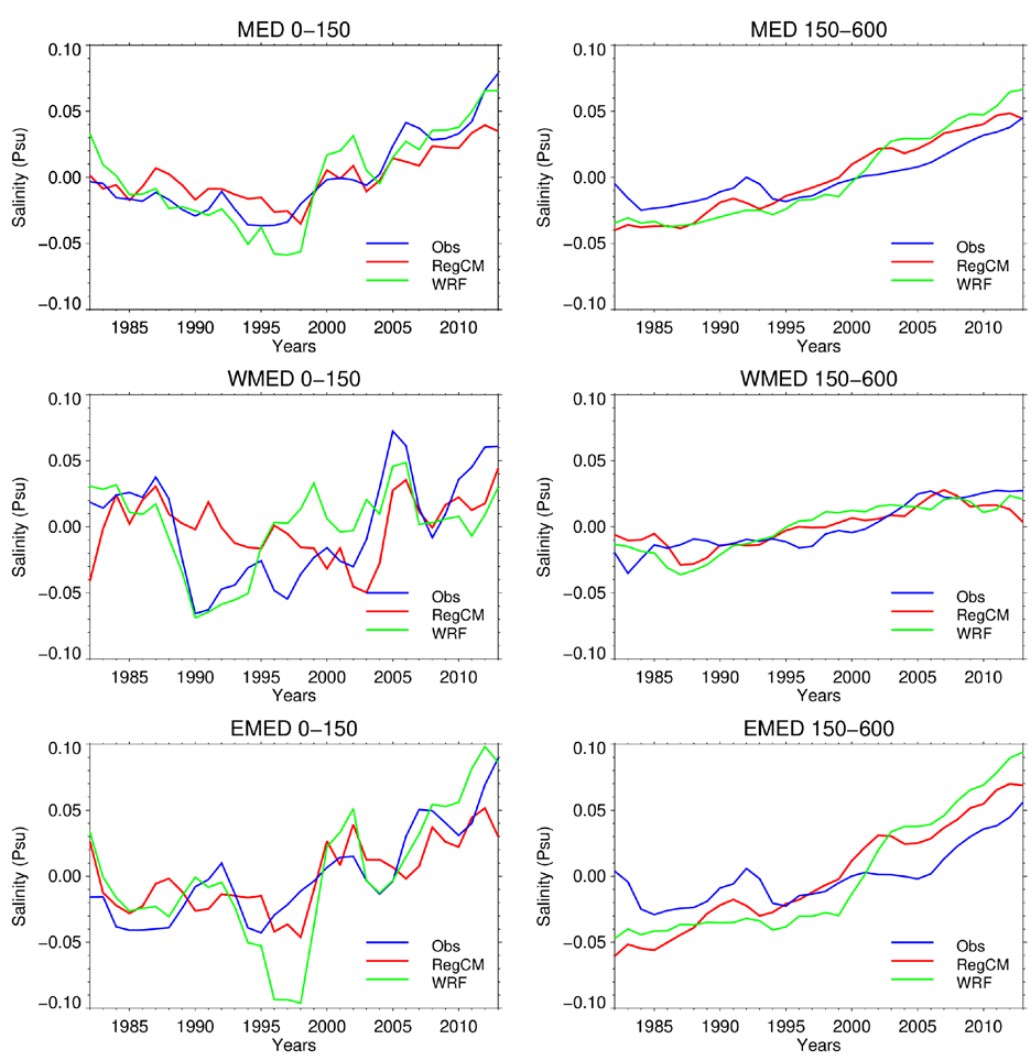

**Figure 16**. Annual mean salinity anomalies (*psu*) for upper (0-150 m) and intermediate (150-600 m) layers of the Mediterranean Sea, Western and Eastern basins over the period 1982-2013.





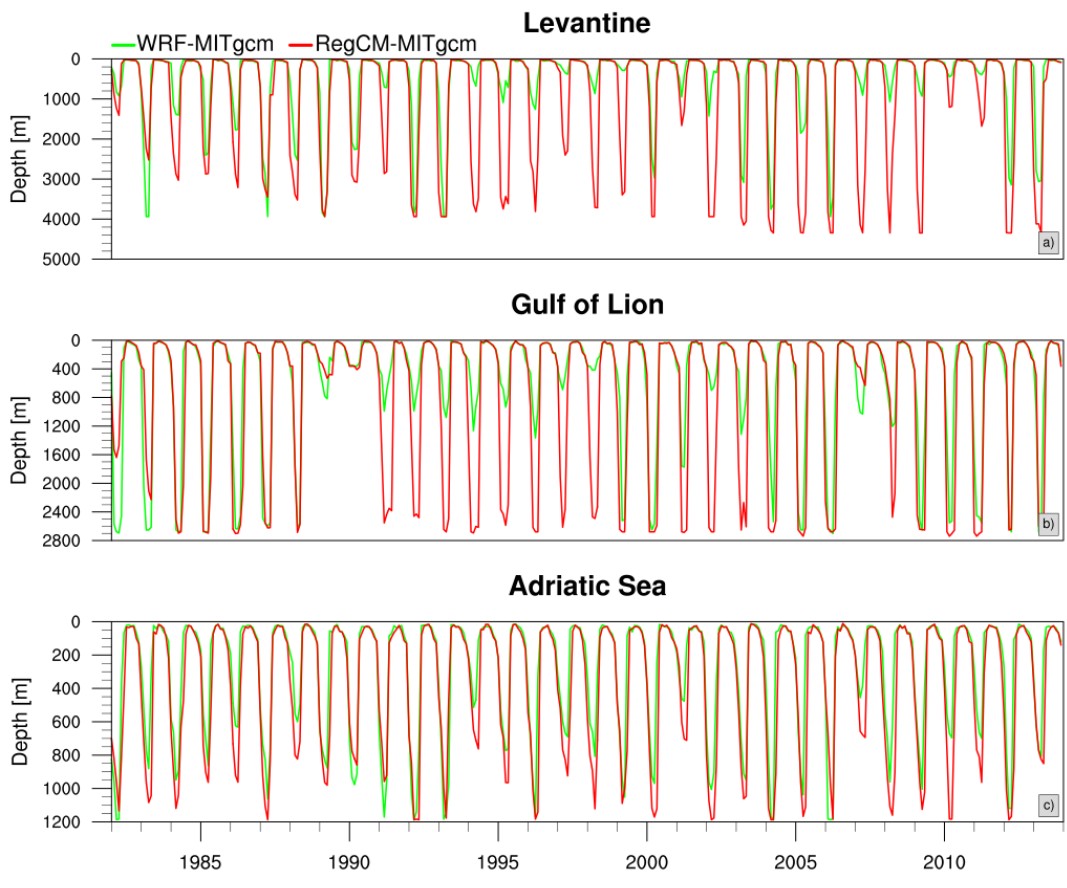

**Figure 17.** Time evolution of the maximum MLD computed over the Levantine basin, Gulf of
Lion area and Adriatic Sea for WRF-MITgcm (green) and RegCM-MITgcm (red) simulations.



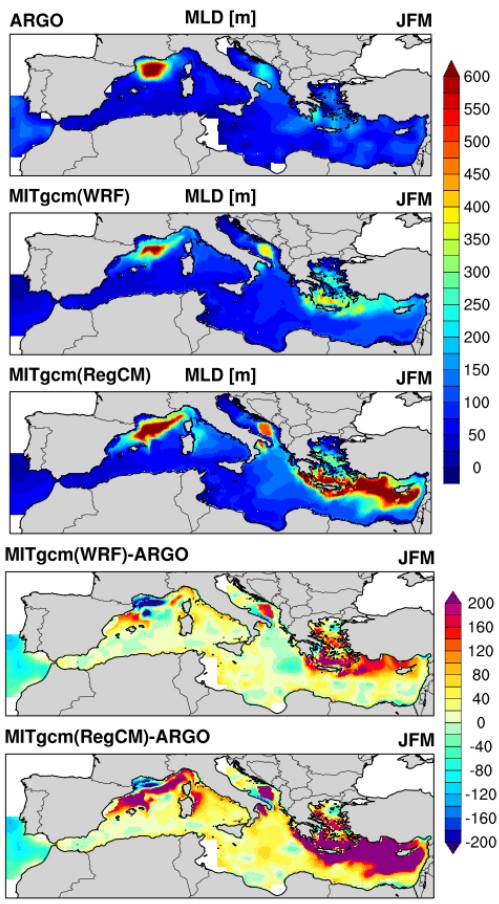


**Figure 18.** Winter (JFM) spatial pattern (upper three panels) and bias (lower two panels) of
mixed layer depth (MLD [$m$]) as simulated by the coupled model using the two atmospheric
components as forcing (i.e. WRF and RegCM) and ARGO dataset between 1982 and 2013. Note
that ARGO data are interpolated into the ocean model grid for comparison purposes.

1312

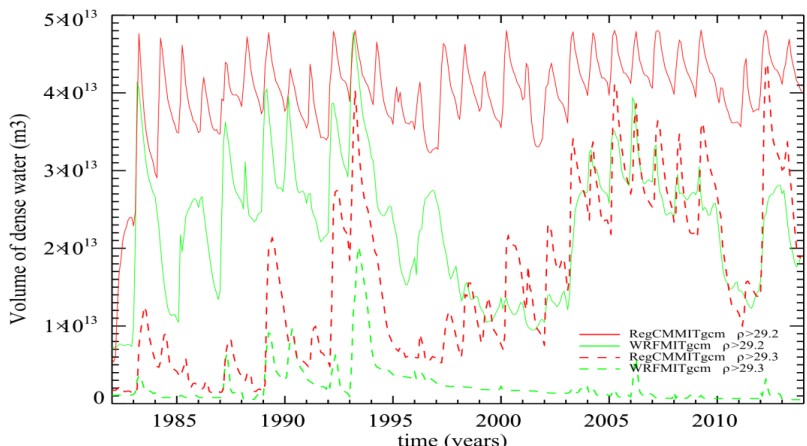

1313  **Figure 19.** Monthly volume of water denser than 29.2 kg m$^{-3}$ (solid line) and denser than 29.3 kg
1314  m$^{-3}$ (dashed line) produced in the Cretan Sea for the two configurations of ENEA-REG system.