# Peer review of "THE ENEA-REG SYSTEM (v1.0), A MULTI-COMPONENT REGIONAL EARTH SYSTEM MODEL. SENSITIVITY TO DIFFERENT ATMOSPHERIC COMPONENTS OVER MED-CORDEX REGION"

_Geoscientific Model Development, 2020_

## Author Response (AR1)

**Response to RC1:**
**We would like to thank the reviewer for the time taken to read and comment on this manuscript. The comments have been very helpful to improve the manuscript. We will follow your suggestions in addressing these changes in the revised version. Please find below (in bold) our responses to the reviewer's comments.**

The manuscript describes and evaluates a new regional Earth System Model (ESM), the ENEA-REG system. The system's main components are a regional ocean model, the MITgcm, a river routing scheme, the HD, and two choices of atmospheric models, the RegCM and the WRF. The description paper for ENEA-REG, version 1.0, is certainly needed, and the GMD is the proper journal for this. But, there are a few issues the authors should work on before the final publication.

It is an advantage to have the possibility to choose between two atmospheric models in the ESM. This possibility allows a, as it is tried in the manuscript, fair comparison and investigation of the impact of different atmospheres on the, e.g., ocean circulation. But, both atmospheric models itself have many options available. For example, WRF can choose from various land surface schemes, microphysics schemes etc. Therefore, there are already many different atmospheric forcings to be got from different WRF setups. Why using two different models, which will, e.g., increase the challenge of future support? This decision should be more strongly motivated in the introduction.

**Certainly regional climate models offer several schemes and parameterizations. However, it is also well recognized that models are flawed and some numerical schemes poorly perform locally or over some regions. For these reasons models with relevant skills in simulating climate in some regions of the World could produce large local biases in other regions. In principle, using a combination of different models allows to overcome this issue as users can select the model to be used depending on the study area and the skills of the model over the region of interest. We have further clarified this into the introduction.**

The discussed simulations were driven by ERA-Interim reanalysis. WRF was nudged, i.e. not only driven at the domain boundaries, to ERA-Interim. It has been shown before that nudging improves the simulation quality, but only if nudged to reanalysis data. It is probably not helpful to nudge against any global climate projection, which is far less good than any reanalysis, as nudging imprints the driving models deficits on the nested simulation (e.g. Leps et al. 2018). And, RegCM was not nudged. I would prefer to see a comparison with both atmospheric components not nudged. Perhaps, the simulation results would be more similar. On the other hand, the different atmospheric results force the ocean differently, which shows the impact of different atmospheres more clearly.

**We agree with the reviewer on comparing not nudged atmospheric components; for this reason we reshaped the manuscript replacing WRF results with those from the non-nudged simulation. Further details can be found in the revised version of the manuscript. The comparison with the nudged simulation has been however discussed in the conclusion paragraph of the revised manuscript.**

I additionally would like to get a bit of information about the computational costs. WRF is more expensive than RegCM? The ocean MITgcm cost is negligible?

**In the present configuration a straightforward comparison of computational costs was not possible as atmospheric models have different horizontal and vertical grids. Anyhow, the**

**atmospheric components are slower than the ocean model because of the larger amount of processes to be simulated (i.e. radiation, clouds, processes taking place at land surface, soil hydrology). In addition, in order to respect the CFL conditions the physical timestep of the atmospheric component is about 10 times smaller compared to the ocean model.**
**Overall, for the present domains the average CPU time for one year of simulation is around 1 day. A detailed benchmark of RegESM performances is given by Turuncoglu (2019).**

**Turuncoglu, U. U.: Toward modular in situ visualization in Earth system models: the regional modeling system RegESM 1.1, Geosci. Model Dev., 12, 233–259, https://doi.org/10.5194/gmd-12-233-2019, 2019.**

The authors often used the reanalysis ERA5 as a reference, e.g., for 10-m wind over the sea. It should be made clear that ERA5 over the Mediterranean Sea might be off too and should be taken cautiously.

**Thanks for pointing this out; we clarified in the text that ERA5 should be used cautiously over the Mediterranean Sea.**

The statement on page 10, line 279, "no single combination of parameterizations yields optimal results" is a bit misleading. This statement is used as an explanation of a temperature bias larger than 4°C, which is quite substantial. It is true that no parameterization, and no combination of parameterizations, can be perfect, but still there are successful global climate simulations. I would avoid using this excuse here.

**We apologize for the misunderstanding: the sentence "no single combination of parameterizations yields optimal results" is not an excuse to explain the large bias found in temperature but the outcome of a different sensitivity study where the authors performed several experiments changing WRF set-up and finding no optimal combination able to remove the winter cold bias in the North-Eastern Europe (i.e. Mooney et al. 2013). We reshaped the sentence to better reflect results of other studies where this bias is analyzed and discussed.**

The language of the paper should be improved. A few examples are:

Titel: ".", perhaps a "-"?

**We improved the language correcting a few typos and errors.**

Abstract: line 21: "remarkable biases are relevant for some variables" -> relevant for processes, seen in simulated values of different variables?

**We reshaped the sentence.**

page 17, line 514: "estimation .... has been faced" -> "the challenge of estimation ... has been faced"

**We changed as suggested.**

page 26, line 785: "climate constraint by coupling" -> the simulated climate is modified by the actively coupled Med. sea or similar. The coupling itself cannot change the climate, and the Med. sea cannot constrain but modify the European climate.

**Thanks for the suggestion, we reshaped the sentence.**

Leps, N., Brauch, J., & Ahrens, B. (2019). Sensitivity of Limited Area Atmospheric Simulations to Lateral Boundary Conditions in Idealized Experiments. *Journal of Advances in Modeling Earth Systems*, *11*(8), 2694–2707. https://doi.org/10.1029/2019MS001625

**Response to RC2:**
**We would like to thank the reviewer for the time taken to read and comment on this manuscript. The comments have been very helpful to improve the manuscript. We will follow your suggestions in addressing these changes in the revised version. Please find below (in bold) our responses to the reviewer's comments.**

The paper introduces a new regional Earth system model for use in the Med-CORDEX region. The main novelty of the model is the possibility of using two different atmospheric components and different land surface schemes. In the validation part of the paper it is shown that the characteristics of the atmospheric model play an important role in the ability of the coupled model to simulate the present time climate and that different ocean biases arise depending on which atmospheric component is used. In general, the paper shows that the new model shows a performance that is comparable to the state of the art regional coupled models that contribute to Med-CORDEX and can be used for climate studies in the region. Therefore, the paper deserves publication, but after the following comments are addressed.

Major

1.  Why there is not an explicit representation of the Black Sea? How do you determine the heat and mass exchange between the Black Sea and the Mediterranean?

    **In the present simulations the Black Sea contribution has been treated as a climatological river. As an alternative, that could be a future improvement of the RegESM, we should enhance the horizontal resolution of the ocean model to solve directly the transport at the Bosphorous/Dardanelles Straits including also the Black Sea in the ocean model simulation domain. We have modified the text and clarified this point.**

2.  Why do you use spectral nudging? One of the adventages of the coupling is the freedom that the regional model has to develop its own, physically consistent climate. You are imposing a strong physical constrain that can be unnecessary, as the large scale climate is never too different from the global model, except when reflect an important issues related to domain size and location (see e. g. Sein et al, 2014). Could you elaborate on the reasons that lead you to use the nudging?

*Dmitry V. Sein, Nikolay V. Koldunov, Joaquim G. Pinto & William Cabos (2014) Sensitivity of simulated regional Arctic climate to the choice of coupled model domain, Tellus A: Dynamic Meteorology and Oceanography, 66:1, DOI: 10.3402/tellusa.v66.23966*

**It is widely known that regional climate models tend to drift away from the driving fields and nudging has been developed to address this issue. Consistent with previous studies (e.g. Liu et al., 2012; Zittis et al., 2018), we consider the nudging an added value, thus we decided to used it to provide to the ocean model the driving field as realistic as possible. However, we agree that**

**comparing models with and without nudging is unfair. For this reason we revised the whole manuscript presenting results from WRF-MITgcm using WRF without nuding.**

*Liu, P., Tsimpidi, A. P., Hu, Y., Stone, B., Russell, A. G., and Nenes, A.: Differences between downscaling with spectral and grid nudging using WRF, Atmos. Chem. Phys., 12, 3601–3610, https://doi.org/10.5194/acp-12-3601-2012, 2012.*

*Zittis, G.; Bruggeman, A.; Hadjinicolaou, P.; Camera, C.; Lelieveld, J. Effects of Meteorology Nudging in Regional Hydroclimatic Simulations of the Eastern Mediterranean. Atmosphere 2018, 9, 470. https://doi.org/10.3390/atmos9120470*

3. I miss a comparison of the coupled runs with an ERA5 forced oceanic simulation. It would help to clarify the contribution of the oceanic formulation to the biases. In particular, to the positive SST biases, which are of opposite sign with other regional coupled models, e. g. the MEd-CORDEX ensemble used in Darmarki et al (2019)

**The suggested comparison is not so straightforward, as in general stand alone ocean simulations use relaxation techniques to the prescribed SST. However, some inferences can be done based on the results of this paper. In the revised version of the paper, that shows the results of the coupled simulations avoiding the use of nudging, the SST biases in the summer season are similar, while this bias is considerably reduced by the application of nudging (see former version of the paper) that keeps the atmospheric fields close to the forcing global simulation.**

4. The short spin-up time can be of relevance to the behaviour of the simulated mixed layer, deep wáter formation and the temperature and salinity in the intermediate layer, as suggested by figure 15 of Parras-Berrocal et al (2020). In general, this figure and considerations of basin size suggest a spin-up time of around 80 years.

**Our choice has been to start the simulation with an ad-hoc initial condition during a month in which the ocean is well stratified. This choice has been proven to be adequate to the purpose of this study, as it is shown in figures 15 and 16, where interannual variability of temperature and salinity in the intermediate layer are well reproduced.**

Minor

Line 39 "the regional climate"

**Changed as suggested.**

Line 56 please, add the following reference:

**Thanks for reporting the missing reference, we have added it.**

Soto-Navarro, J., Jordá, G., Amores, A. et al. Evolution of Mediterranean Sea water properties under climate change scenarios in the Med-CORDEX ensemble. Clim Dyn 54, 2135–2165 (2020). https://doi.org/10.1007/s00382-019-05105-4

Line 61. Would be better "A number of model studies"  instead  of "Future model projections"

**Changed as suggested.**

Line 69: " we evaluate the ability of of the ENEA-REG system to represent adequately the present climate of the Mediterranean by" instead of "perform the evaluation run of the ENEA-REG system"

**Changed as suggested.**

Line 77 differing in

**Changed as suggested.**

Line 85 correct "applications.For"

**Typo, thanks for reporting.**

Line 95 please deleto "to glue", does not sound fit for the text

**We removed "to glue".**

Lin 112 "in the experiments"

**Got it.**

Line 123 "can be run with two" instead of "is made up of two"

**We prefer to keep the original sentence as "can be run with two interchangeable atmospheric components " could result confusing for some readers.**

Line 129  "For any region"

**Thanks for suggesting.**

Line 132 Any reason for using these parameterizations? Have you tuned the model in coupled mode?

**Yes, we tested the coupled model with several different configurations, finding that surface wind speed is the key variable for the coupling: in case of poor performances in reproducing 10m winds, the ocean model is not able to reproduce the correct surface patterns and Mediterranean circulation. Among the tested configurations, the present set-up produces good results while preserving an  efficient computational time.**

Line 187  coupled to a global atmosphere? Or as the oceanic component of a global coupled model?

**We have reshaped the phrase. Anyhow, the MITgcm has been used both coupled to a global atmosphere (e.g. Polkova et al., 2014) and as the oceanic component of a regional coupled model (Artale et al., 2010; Sun et al.,2019).**

Line 205. Still, the spinup would be useful.  Asstrssed above and also shown in Soto-Navarro et al, a short spinup or its absence can have a strong impact on the  simulation, especially in the deeper layers.

**We fully agree that for climate studies long spin-up are desirable; however, as already stated in the manuscript, the aim of this study is the comparison of two coupled model systems having in common the same ocean model. As the MITgcm has the same initial and boundary conditions in its two configurations the differences in results are only due to differences in surface forcing.**

Line 222. Outside the regional model  domain, does the forcing come from ERA-Interim?

**In case of atmospheric components forcing data come from ERA-Interim, while, considering the MITgcm, temperature and salinity boundary conditions in the Atlantic Ocean come from the global LEVITUS94 climatological monthly 3D data.**

Line 232 Does not the short spin up period influence the  simulated mixed layer and especially the Deep w áter formation? How Good is the simulation of the Nile discharge?

**As already stated before, the limited spin-up period did not affect the results of MLD and deep water formation.**
**The Nile river discharge has not  been simulated, as the whole catchment basin is not covered by the domains of atmospheric models. In addition, the discharge computed by the river routing model would be very different compared to the observed discharge as the model does not consider the anthropic use and regulation of freshwater. For these reasons we decided to prescribe Nile discharge as a climatological boundary condition. This point has been now clarified in the text.**

Line 250. ERA5 is a reanalysis and is not directly based on observational data. Why do not use a regional reanalysis for validation (e.g. https://climate.copernicus.eu/copernicus-regional-reanalysis-europe-cerra) in order to evaluate the simulation of the climate fields on smaller scales?

**This is a good point we already faced at the beginning of the analysis. Although for different variables several observational-based dataset exist, we decided to use the same reference dataset (i.e. ERA5) to keep the validation consistent for each atmospheric variable. In addition, ERA5 is similar to ERA-Interim, thus such comparison allows to assess how much our regional models drift from large scale forcing.**

Line 263: The maximum and minimum daily temperature could show better the impact of the parameterizations on temperature

**Thank you very much for the suggestion; we have also considered the timeseries of daily maximum and minimum temperature, especially in the zone of the domain where the bias of the temperature is more important. However, as already discussed in the text, this is a well known problem in WRF model possibly related to soil physics, surface layer transfer and PBL scheme. Finally, we remark that this large bias does not affect the results over the Mediterranean region which is the focus of the current paper.**

Line 277. Is this true for the uncoupled or the coupled mode?

**The chosen settings reproduce at the best the wind field in both coupled and stand-alone simulations.**

Line 400  what about cloud cover?

**Thank you for the observation, we have accordingly modified the text.**

Figure 3: Two different colorscales for biases difficult their comparison. Please, correct

**We have changed the colorscales of Figure 3.**